# Distinct RanBP1 nuclear export and cargo dissociation mechanisms between fungi and animals

Yuling Li[1†], Jinhan Zhou[1†], Sui Min[1†], Yang Zhang[2], Yuqing Zhang[1], Qiao Zhou[1], Xiaofei Shen[3], Da Jia[3], Junhong Han[2], Qingxiang Sun[1*]

[1]Department of Pathology, State Key Laboratory of Biotherapy and Cancer Center, West China Hospital, Sichuan University, Collaborative Innovation Centre of Biotherapy, Chengdu, China; [2]Division of Abdominal Cancer, State Key Laboratory of Biotherapy and Cancer Center, West China Hospital, Sichuan University, Collaborative Innovation Centre for Biotherapy, Chengdu, China; [3]Key Laboratory of Birth Defects and Related Diseases of Women and Children, Department of Paediatrics, Division of Neurology, West China Second University Hospital, Sichuan University, Chengdu, China

**Abstract** Ran binding protein 1 (RanBP1) is a cytoplasmic-enriched and nuclear-cytoplasmic shuttling protein, playing important roles in nuclear transport. Much of what we know about RanBP1 is learned from fungi. Intrigued by the long-standing paradox of harboring an extra NES in animal RanBP1, we discovered utterly unexpected cargo dissociation and nuclear export mechanisms for animal RanBP1. In contrast to CRM1-RanGTP sequestration mechanism of cargo dissociation in fungi, animal RanBP1 solely sequestered RanGTP from nuclear export complexes. In fungi, RanBP1, CRM1 and RanGTP formed a 1:1:1 nuclear export complex; in contrast, animal RanBP1, CRM1 and RanGTP formed a 1:1:2 nuclear export complex. The key feature for the two mechanistic changes from fungi to animals was the loss of affinity between RanBP1-RanGTP and CRM1, since residues mediating their interaction in fungi were not conserved in animals. The biological significances of these different mechanisms in fungi and animals were also studied.
DOI: https://doi.org/10.7554/eLife.41331.001

*For correspondence:
sunqingxiang@hotmail.com

[†]These authors contributed equally to this work

Competing interests: The authors declare that no competing interests exist.

## Introduction

Eukaryotic cells each have a nucleus which segregates the nucleoplasm and cytoplasm into two isolated compartments. Exchanges between these compartments are mainly mediated through nuclear pore complex (NPC), a semi-permeable channel that allows only certain classes of molecules to pass through, for example importins and exportins, which are collectively called karyopherin proteins (*Beck and Hurt, 2017*). Cargo entering the nucleus must possess a nuclear localization signal (NLS), which binds to an importin and enters nucleus through NPC (*Rexach and Blobel, 1995*). In the nucleus, the GTP-bound form of Ran (Ras-related nuclear) protein dissociates the importin-NLS cargo, and RanGTP-Importin is exported to cytoplasm (*Izaurralde et al., 1997*; *Kutay et al., 1997*). For a cargo's nuclear export, its nuclear export signal (NES) forms a complex with exportin in the presence of RanGTP, and together the trimeric complex translocates to the cytoplasm through NPC (*Ullman et al., 1997*). In the cytoplasm, RanGTP complexes (with either importin or exportin-NES) are hydrolyzed to RanGDP by GTPase-activating protein RanGAP, dissembling the complexes and recycling karyopherins and Ran for further rounds of nuclear transport (*Bischoff et al., 1995a*).

However, importin or exportin-NES displays extremely high affinity (in the nM range) for RanGTP and inhibits RanGAP-facilitated RanGTP hydrolysis (*Bischoff and Görlich, 1997*; *Askjaer et al.,*

**eLife digest** Plant, animal and fungal cells all store their DNA inside the cell's nucleus. Small molecules can freely cross the membrane that surrounds the nucleus, but pores in the membrane control when larger molecules enter or leave. This transport process is an essential part of healthy cell behavior.

To leave the nucleus, large molecules need to carry a coded sequence called a nuclear export signal. In yeast cells, which are often used to study cell biology, this sequence allows cargo to bind to a groove in so-called molecular cargo vehicles, such as a protein called CRM1. A protein called RanGTP binds to CRM1 to supply the energy needed to transport molecules across the membrane. Outside of the nucleus, another protein called RanBP1 closes up the groove in the CRM1 protein to help remove the cargo by interacting with RanGTP and CRM1 to form a 'complex'.

The version of RanBP1 found in animal cells has its own nuclear export signal, which led researchers to question whether it works in the same way as yeast RanBP1. To find out, Li et al. compared yeast RanBP1 with mouse RanBP1. This revealed that mouse RanBP1 lacks the amino acids that allow it to interact with CRM1 in the fashion of yeast RanBP1. When unloading cargo from CRM1, mouse RanBP1 does not form a complex with Ran and CRM1; instead, it works entirely through removing RanGTP from CRM1. This process is more efficient than the one used by yeast cells, but it uses twice as much energy.

The results presented by Li et al. demonstrate that even processes that are essential to cells can be optimized to fit the needs of different species. Future work could potentially exploit the differences in the export processes used by fungi and animal cells to develop new anti-fungal treatments.

DOI: https://doi.org/10.7554/eLife.41331.002

*1999*). Efficient hydrolysis requires Ran binding proteins containing one or more Ran binding domains (RBDs, around 150 residues each) to dissociate RanGTP from karyopherin prior to hydrolysis (*Beddow et al., 1995*; *Bischoff et al., 1995b*; *Vetter et al., 1999*). In human, there are two such proteins, namely RanBP1 and RanBP2. While the predominantly cytoplasmic RanBP1 contains one RBD, the cytoplasmic rim-attached RanBP2 has four RBDs (*Bischoff et al., 1995b*). These RBDs are the tightest binders of RanGTP, $K_d$ being around 1 nM, whereas it binds to RanGDP at only approximately 10 µM affinity (*Görlich et al., 1996*; *Kuhlmann et al., 1997*; *Delphin et al., 1997*).

In addition, RanBP1 in the cytoplasm functions in the disassembly of nuclear export complexes (also before RanGTP is hydrolyzed), through dissociating NES containing cargoes from the complexes (*Floer and Blobel, 1999*). CRM1 (Chromosomal Region Maintenance 1, also known as Exportin-1) is a major nuclear export factor that is responsible for nuclear export of a plethora of proteins containing NES sequence(s) (*Stade et al., 1997*). In yeast (*S. cerevisiae*), after binding to RanGTP and CRM1, RanBP1 allosterically closes the groove, releasing CRM1 cargoes into the cytoplasm (*Koyama and Matsuura, 2010*).

Sequence analysis indicates that in contrast with fungi RanBP1 (which includes yeast RanBP1), animal RanBP1 contains NES sequence C-terminal to its RBD (*Zolotukhin and Felber, 1997*) (*Figure 1— figure supplement 1*). It is reported that the NES of human RanBP1 (hRanBP1) is responsible for its cytoplasmic accumulation (RanBP1 is a shuttling protein) (*Zolotukhin and Felber, 1997*; *Künzler et al., 2000*). If animal RanBP1 binds to RanGTP-CRM1 similarly as yeast RanBP1 (yRanBP1), an apparent paradox then exists: its NES binding to CRM1 is inhibited by its own RBD. Specifically, if animal RanBP1 binds to CRM1 through NES in the nucleus to prepare for nuclear export, its RBD might immediately interact with RanGTP on CRM1 (because of proximity and high affinity) and dissociate its own NES before animal RanBP1 is exported. One may argue that NES may play a role in recruiting animal RanBP1 to CRM1. However, RanBP1-RanGTP-CRM1 complex displayed much higher affinity than NES-CRM1-RanGTP complex in yeast (*Maurer et al., 2001*); thus, the recruiting purpose seems unnecessary and unlikely. It should be noted that there are about ten residues between RBD and NES, which are insufficient to cover the distance (about 70 Å) between RBD and NES in space. Therefore in theory, the NES and RBD of animal RanBP1 would not bind to CRM1-RanGTP simultaneously. It is fascinating as to why animal RanBP1 requires an extra NES while fungi

RanBP1 does not; what factor(s) prevents animal RBD from dissociating its own NES during its nuclear export; whether the NES of animal RanBP1 functions in cargo dissociation by direct competition with NES of cargo; and how animal RBD and NES binding to CRM1-RanGTP is regulated in time and space in cells. Intrigued by these long-standing questions, we performed biochemical, biophysical, and cellular studies on RanBP1 and related proteins. Our work not only solved those puzzles, but also discovered unexpected animal RanBP1 nuclear export and cargo dissociation mechanisms distinctive from those in the yeast.

## Results

### Mouse and yeast RanBP1 bind to CRM1-RanGTP differently

In yeast, RanBP1, RanGTP and CRM1 form a tight complex whereby RanBP1 forces H9 loop of CRM1 to allosterically close NES binding groove and dissociate NES. In order to visualize the mode of animal RanBP1's binding to RanGTP-CRM1, we first attempted to solve the crystal structure of complex formed by mouse RanBP1 (mRanBP1), human RanGTP (hRanGTP) and human CRM1 (hCRM1). However, although the yeast complex formed readily as expected, the equivalent complex with animal proteins hardly formed under similar conditions (*Figure 1A*). The human and yeast Ran protein shares 83% sequence identity and are often used interchangeably (*Koyama and Matsuura, 2010*; *Sun et al., 2013*). When RanBP1, RanGTP and CRM1 were mixed at 5:3:1 molar ratio and passed through a size exclusion chromatography column, the yeast proteins formed stable complex and were co-eluted, but not the case for the animal proteins (*Figure 1—figure supplement 2*). Interestingly, a greater amount of animal complex was formed when concentration of RanGTP was increased (*Figure 1B*). In contrast, the yeast complex was not affected by RanGTP concentration (*Figure 1B*). It should be noted that the mammalian complex was dependent upon high RanGTP but not RanGDP concentration (*Figure 1—figure supplement 3*). These results suggest that mRanBP1 is somewhat different from yRanBP1, in forming complex with RanGTP-CRM1.

### Mouse RanBP1's NES is necessary for CRM1 binding and its nuclear export

To identify which region of mRanBP1 mediated the interaction with hCRM1, we cloned GST-tagged mRanBP1 mutants with NES mutation (termed as NESmut), the C terminus deletion (ΔC) or only the NES (NES$^{mRanBP1}$). Surprisingly, both NESmut and ΔC lost binding to hCRM1 completely, in either low or high RanGTP concentration (*Figure 1C*). In contrast, NES$^{mRanBP1}$ bound to hCRM1 in the presence of RanGTP, and was outcompeted by supraphysiological NES (*Engelsma et al., 2008*) from the Minute Virus of Mice (*Figure 1D*), suggesting that NES$^{mRanBP1}$ is a regular NES that binds to NES groove on hCRM1. Further, the NES of mRanBP1 is weaker than that of regular NES from PKI (Protein kinase inhibitor) (*Figure 1E*). Previously, it was reported that the NES of mRanBP1 was essential for its nuclear export (*Plafker and Macara, 2000*). Indeed, while mRanBP1 was localized to the cytoplasm, NESmut (or mRanBP1 in the presence of CRM1 inhibitor KPT-330) was significantly re-localized to the nucleus (*Figure 1F*). Unexpectedly, yRanBP1 was exclusively localized in the nucleus (*Figure 1F*, right column), though RanBP1 proteins are known to be cytoplasmic localized proteins (*Künzler et al., 2000*; *Plafker and Macara, 2000*). The reason for this counter-intuitive observation will be explained in later sections. Altogether, these results conclude that the NES of mRanBP1 is necessary and sufficient for its interaction with hCRM1 in the presence of RanGTP, and this interaction is crucial for nuclear export of mRanBP1.

### Mouse RBD dissociates cargo through sequestering RanGTP

Previously we showed that ΔC or NESmut did not bind to hCRM1 in the presence of RanGTP (*Figure 1C*). Relating to RanBP1's function of CRM1 cargo dissociation, we wondered whether either ΔC or NESmut possesses cargo dissociation ability, and whether NES$^{mRanBP1}$ is involved in dissociating cargo, possibly through direct competition with cargo's NES. Pull down assay showed that ΔC or NESmut dissociated cargoes as potently as WT mRanBP1 (*Figure 2A*, *Figure 2—figure supplement 1*), suggesting that NES of mRanBP1 is dispensable for cargo dissociation. Further, if direct competition plays a role, the NES of mRanBP1 should be stronger than that of the cargo. However, our

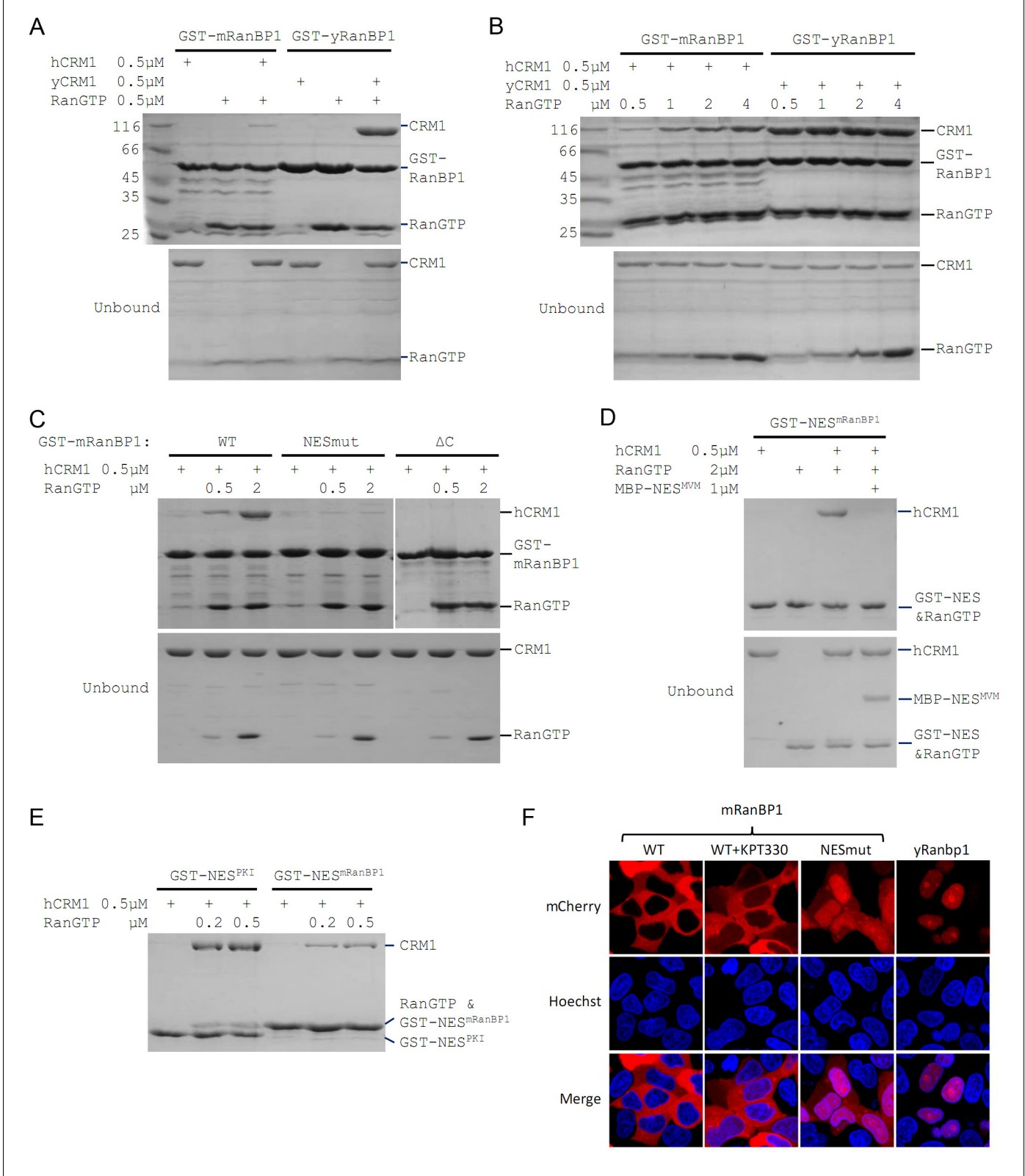

**Figure 1.** Mouse RanBP1 forms a nuclear export competent complex with hCRM1 only through its NES, and in a RanGTP concentration dependant manner. (A, B, C) GST tagged mouse/yeast RanBP1 or mutants pull down of CRM1 in the presence of RanGTP. (A) While yeast proteins form a strong RanBP1-RanGTP-CRM1 complex, the animal proteins form a RanBP1-RanGTP complex with very weakly bound CRM1. (B) At increasing concentration of RanGTP, while yeast complex is not affected, animal CRM1 is increasingly bound to GST-mRanBP1-RanGTP. (C) Mutation (NESmut) or deletion of the

*Figure 1 continued*

NES of mRanBP1 (ΔC) abolishes hCRM1 binding at either high or low RanGTP concentration. (D) GST-NES^mRanBP1 pull down of CRM1 in the presence of RanGTP and NES inhibitor MBP-NES^MVM. NES of mRanBP1 binds to hCRM1 in the presence of RanGTP and is effectively outcompeted by MBP-NES^MVM, suggesting that the NES of mRanBP1 binds to NES groove of CRM1. (E) GST-NES pull down of hCRM1 and different concentration of RanGTP to show that the NES of mRanBP1 is weaker than the well-studied NES of PKI cargo. (F) mCherry tagged different RanBP1 constructs were transfected into HeLa cells and treated with or without 5 μM CRM1 inhibitor KPT-330 for 3 hr. While mRanBP1 is exclusively cytoplasmic, mRanBP1 treatment with KPT-330 and NESmut are significantly re-localized to the nucleus. yRanBP1 is exclusively nuclear.

DOI: https://doi.org/10.7554/eLife.41331.003

The following figure supplements are available for figure 1:

**Figure supplement 1.** NES of RanBP1 is conserved in animals but does not exist in fungi.

DOI: https://doi.org/10.7554/eLife.41331.004

**Figure supplement 2.** In excess of RanBP1, mRanBP1, hRanGTP and hCRM1 do not form complex like yeast proteins.

DOI: https://doi.org/10.7554/eLife.41331.005

**Figure supplement 3.** GST-mRanBP1 pull down of hCRM1 in the presence of RanGTP or RanGDP.

DOI: https://doi.org/10.7554/eLife.41331.006

**Figure supplement 4.** Structure based sequence alignment of human, mouse, zebrafish, frog and yeast RanBP1 and 4 RBDs from human RanBP2.

DOI: https://doi.org/10.7554/eLife.41331.007

earlier result showed that NES^mRanBP1 was weaker than cargo PKI's NES (*Figure 1E*). Taken together, we conclude that mRanBP1 does not dissociate cargo through direct competition of cargo's NES.

Since ΔC-RanGTP did not bind to hCRM1 but still dissociated cargo (*Figures 1C* and *2A*), we therefore speculated that ΔC dissociates cargo through sequestering RanGTP from the export complex, which would result in a transient, low-affinity binary complex of hCRM1 and NES cargo that automatically dissociates. Indeed, at high concentration of RanGTP, cargo dissociation activity of ΔC was fully inhibited (*Figure 2B*), because excess of RanGTP not only saturated (and inhibited) mRanBP1, but also allowed CRM1 to form complex with GST-PKI. Further, purified 1:1 complex of ΔC-RanGTP, where RBD was already saturated with RanGTP, was unable to dissociate cargo (*Figure 2C*). These results illustrate that the Ran binding domain, but not NES^mRanBP1, is required for CRM1 cargo dissociation, through stripping RanGTP out of the nuclear export complexes.

## Cargo dissociation mechanisms in yeast and animals

We also performed the above cargo dissociation experiments using yeast protein in parallel. In contrast to mRanBP1, cargo dissociation ability of yRanBP1 was not inhibited by excess of Ran, and yRanBP1-yRanGTP remained active to dissociate CRM1 cargo (*Figure 2B and C*). Further, when RanGTP is in large molar excess than CRM1, yRanBP1 dissociated cargo as long as its concentration was higher than CRM1 concentration; in contrast, mRanBP1 dissociated cargo only at concentration that was higher than that of RanGTP (*Figure 2D*). It is evident that yeast RanBP1 dissociates cargo through sequestering CRM1-RanGTP, distinctive from animal RanBP1's cargo dissociation mechanism (*Figure 3A*).

Previously, it was reported that CRM1 or more generally karyopherins protect bound RanGTP from GAP mediated hydrolysis (*Bischoff and Görlich, 1997*; *Lounsbury and Macara, 1997*). We showed that indeed RanGTP hydrolysis is inhibited when complexed to CRM1 and NES (in the absence of RanBP1), both in yeast and in human. Since human proteins do not form tight trimeric CRM1-RanGTP-RanBP1 complex, it is predicted that hCRM1 would not protect RanGTP-RanBP1 from GAP hydrolysis like the yeast proteins. As expected, the rate of hydrolysis in yeast was partially inhibited with addition of yCRM1, while there was no change of hydrolysis rate with addition of hCRM1 (*Figure 3B*). Using semi-permeabilized cells, we further showed that when CRM1 is the rate limiting factor, nuclear export of cargo is indeed faster in the presence of human CRM1/Ran/RanBP1 than yeast proteins (*Figure 3C,D*), possibly due to faster rate of CRM1 recycling (by faster rate of GAP hydrolysis), although it does not rule out other possibilities (will be discussed later). These experiments are consistent with earlier reports that RanBP1 is required for RanGAP activity, and consistent with our model that animal CRM1/Ran/RanBP1 does not form high-affinity trimeric complex as in yeast.

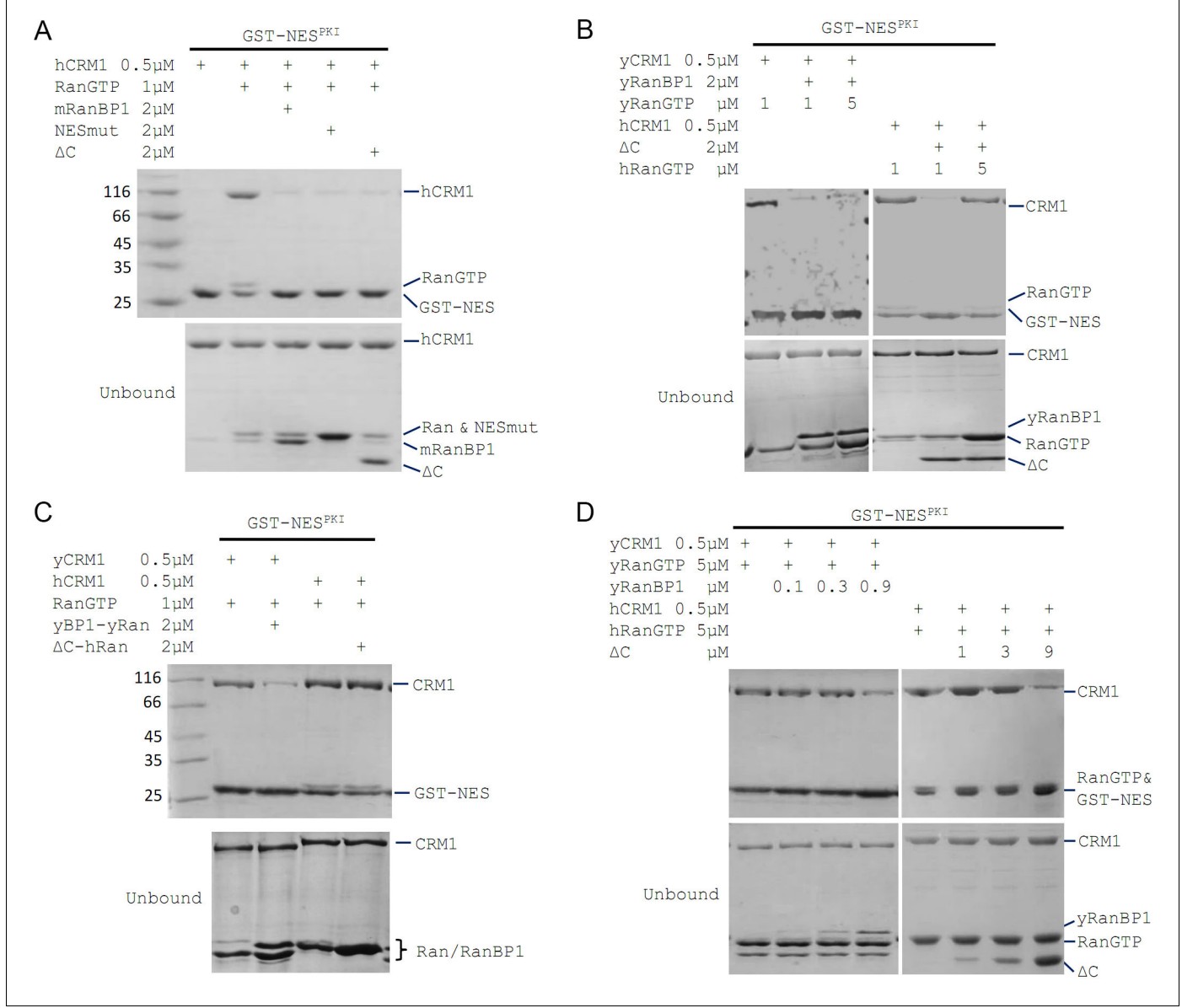

**Figure 2.** mRanBP1's RBD dissociates CRM1 cargo through sequestering RanGTP. (A–D) GST-NES$^{PKI}$ pull down of CRM1 and RanGTP in the presence of RanBP1 or its mutants. (A) NESmut or ΔC dissociates cargo as WT RanBP1. (B) High concentration of RanGTP inhibits cargo dissociation in animals but not yeast. (C) RanGTP-bound ΔC loses its ability in cargo dissociation while yeast RanGTP-RanBP1 remains competent. (D) ΔC dissociates cargo when its concentration is higher than that of RanGTP, and yRanBP1 dissociates cargo when its concentration is higher than that of CRM1.

DOI: https://doi.org/10.7554/eLife.41331.008

The following figure supplement is available for figure 2:

**Figure supplement 1.** GST-NES pull down shows that NESmut or ΔC is as potent as WT RanBP1 in dissociating CRM1 complex.

DOI: https://doi.org/10.7554/eLife.41331.009

## RanBP1 forms complex with CRM1 and two RanGTP proteins in animals

Previously, we showed that mRanBP1 bound to hCRM1 through its NES and that ΔC did not dissociate cargo in excess of RanGTP. Since ΔC forms an extremely tight complex with RanGTP (*Figure 1C*), we hypothesized that when RanGTP is excessive, mRanBP1 binds to one RanGTP through its RBD and binds to hCRM1 through its NES, while hCRM1 simultaneously binds to another RanGTP, forming a tetrameric complex (RanGTP-mRanBP1)-hCRM1-RanGTP (*Figure 4A*). In this model, unlike the yRanBP1 that contacts both yCRM1 and yRanGTP, animal RanBP1's RBD only

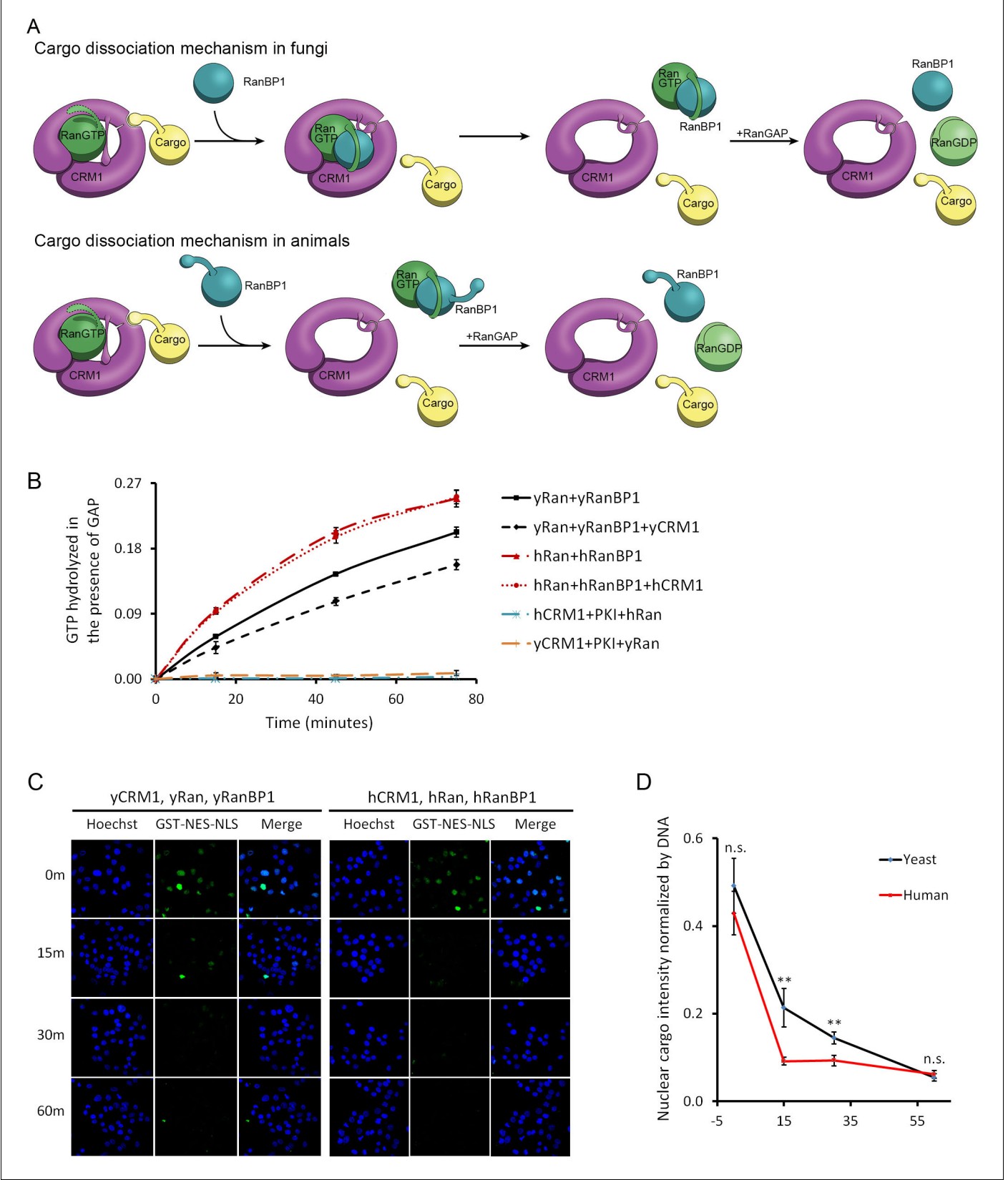

**Figure 3.** CRM1 cargo dissociation mechanism in fungi and animals. (**A**) In fungi, cargoes are actively released from CRM1's NES groove by formation of RanBP1-RanGTP-CRM1 trimeric complex. RanGTP-RanBP1 transiently released from CRM1 is catalyzed by RanGAP. In animals, RanBP1 strips RanGTP

*Figure 3 continued on next page*

*Figure 3 continued*

from CRM1, resulting in dissociation of NES cargo from CRM1. RanGAP catalyzes further dissociation of Ran-RanBP1. (B) GAP mediated RanGTP hydrolysis (arbitrary unit) at different time points. CRM1 and NES (PKI) inhibit RanGTP hydrolysis in both human (cyan) and yeast (orange). Addition of RanBP1 increases the rate of GAP hydrolysis. While addition of yCRM1 partially inhibit GAP mediated hydrolysis, addition of hCRM1 does not. Error bars represent standard deviation of quadruple repeats. (C) In vitro nuclear export of GST cargo in the presence of human or yeast CRM1/Ran/RanBP1 using semi-permeabilized HeLa cells. The reactions were stopped at different time points and level of nuclear cargo was stained with anti-GST antibody. (D) Quantification and statistical analysis of nuclear cargo intensity after normalization by DNA intensity. Error bars represent standard error of measurements for each set of data containing measurements from at least 30 cells. ** denotes p<0.01.
DOI: https://doi.org/10.7554/eLife.41331.010

The following source data and figure supplement are available for figure 3:

**Source data 1.** Nuclear export in the presence of human or yeast proteins using semi-permeablized cells.
DOI: https://doi.org/10.7554/eLife.41331.012

**Figure supplement 1.** Low concentration of Triton-X prevents non-specific cytoplasmic binding and does not permeate the nuclear envelope.
DOI: https://doi.org/10.7554/eLife.41331.011

binds to RanGTP (but not CRM1). Further, in contrast to H9 loop stabilized closure of NES groove in yeast (*Koyama and Matsuura, 2010*), animal CRM1's H9 loop would be shifted away from the NES groove (as in pdb 3NC0) (*Fung and Chook, 2014*), opening its NES groove for interaction with NES of RanBP1 (*Figure 4A*). This model is consistent with all previous results, including that the binding between mRanBP1 and hCRM1 requires excessive RanGTP (*Figure 1A,B*), that NES is the only interacting site between mRanBP1 and hCRM1 (*Figure 1C,D,F*), and that ΔC-RanGTP does not dissociate CRM1 cargo (*Figure 2*).

To further validate this model, we analyzed the size exclusion peaks of yeast and animal RanBP1-RanGTP-CRM1 complexes mixed at a 2:5:1 molar ratio. The calculated RanGTP to RanBP1 molar ratio in the peak of the animal complex was approximately twice (2.05 fold) over that of the yeast complex (*Figure 4—figure supplement 1*), suggesting that animal complex contains two RanGTP proteins. To differentiate the two RanGTP proteins in the tetramer, we performed a pull down assay using GST-RanGTP and a truncated form of Ran, Ran$^{1-179}$, which binds to CRM1 but does not bind to mRanBP1 due to lack of C-terminus. Clearly, Ran$^{1-179}$ and hCRM1 bound to GST-RanGTP beads in the presence of mRanBP1 but not ΔC or buffer (*Figure 4B*). The bound Ran$^{1-179}$ can be explained by the formation of the proposed tetrameric complex, where GST-RanGTP binds to mRanBP1, which further binds to hCRM1-Ran$^{1-179}$ through the NES of mRanBP1. Though ΔC binds to GST-RanGTP, the lack of NES rendered it unable to bind CRM1; therefore there is a lack of Ran$^{1-179}$ band in the bound fraction. By size exclusion chromatography, hCRM1, RanGTP, mRanBP1 and Ran$^{1-179}$ formed a reasonable 1:1:1:1 complex, whereas Ran$^{1-179}$ did not co-elute with CRM1, RanGTP and RanBP1 in yeast (*Figure 4C and D*). Further, RanGTP titration into hCRM1 and mRanBP1 by ITC produced an exothermic phase and an endothermic phase, representing the heat change from forming RanGTP-mRanBP1 and the tetrameric complex, respectively (*Figure 4E*). In contrast, the titration of RanGTP into yCRM1 and yRanBP1 displayed merely an exothermic phase representing the formation of the trimeric complex (*Figure 4F*). Taken together, we conclude that RanBP1 forms a tetrameric complex with CRM1 and two RanGTP proteins in animals (when RanGTP excessive), where RanBP1-RanGTP binds to CRM1-RanGTP through the NES of RanBP1.

## Detection of RanBP1 tetramer in the nucleus

Since the RanBP1 tetrameric complex forms only in excess of RanGTP, and since the nucleus is enriched with RanGTP (*Izaurralde et al., 1997*), we tried to detect the presence of nuclear tetramer using the bimolecular fluorescence complementation (BiFC) approach. We fused one Ran with CYFP (C-terminal fragment of YFP) and another Ran with NYFP (N-terminal fragment of YFP), and co-transfected the two plasmids into 293 T cells. The formation of the tetrameric complex would bring the fluorescent fragments within proximity, promoting the assembly of an intact YFP (*Figure 5A*). Please note that this assay would not differentiate which Ran fusion binds to RanBP1 (or CRM1) in the tetramer, and possibly that both configurations could elicit YFP signals. GST-mRanBP1 pull down showed that these fusion Ran proteins are still functional (*Figure 5—figure supplement 1*). When the fusion plasmids were co-transfected, weak YFP signals were observed (*Figure 5B* yellow arrows), suggesting possible tetramer formation. In fact, not all double fusion transfected cells emitted weak signals,

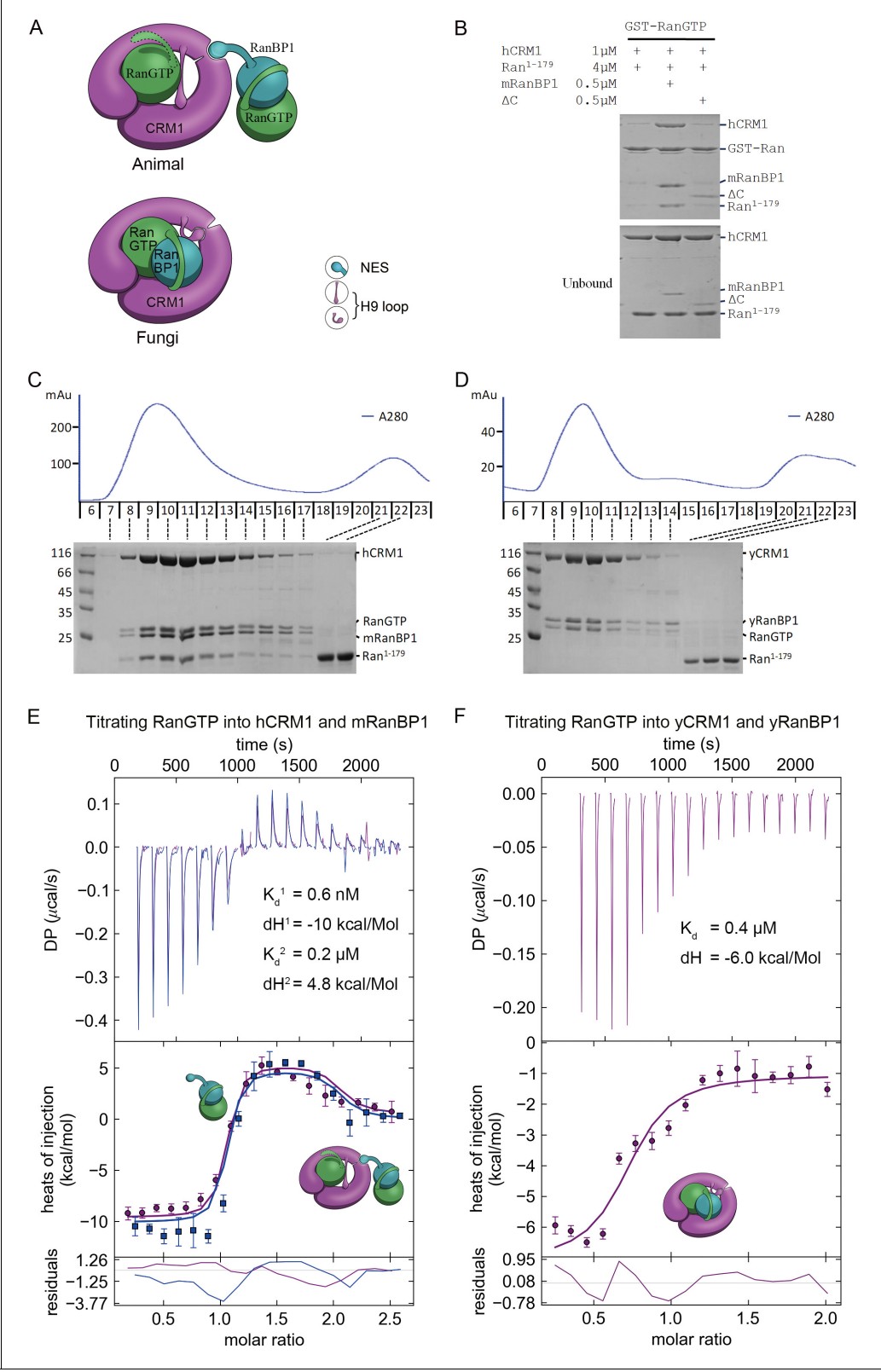

**Figure 4.** Animal RanBP1, RanGTP and CRM1 form a tetrameric complex containing two RanGTP proteins. (**A**) Model of animal and fungi RanBP1 nuclear export complexes. Yeast complex contains one RanGTP and animal complex contains two. Unlike yeast, animal RBD does not contact CRM1 and animal CRM1 has different conformation of NES groove and H9 loop. (**B**) GST-RanGTP pull down of Ran$^{1-179}$ and CRM1 in the presence of mRanBP1, ΔC or buffer. Ran$^{1-179}$ is bound when WT mRanBP1 is added, but not when ΔC is added. Size exclusion profiles and SDS-PAGE analysis of

*Figure 4 continued on next page*

*Figure 4 continued*

the peaks by animal (C) and yeast (D) RanBP1, RanGTP, RanGTP$^{1-179}$ and CRM1 proteins mixed at 1:1:3:1 molar ratio. The first peak contains four bands for animal sample and three bands for yeast samples. ITC titration of RanGTP into animal (E) and yeast (F) CRM1-RanBP1 respectively. Figure E and F include two and one independent titration experiments, respectively. Global or single fit of $K_d$ and $\Delta H$ is displayed in the figure respectively. Error bars represents 95% confidence interval of measurements. While animal proteins display an exothermic phase and an endothermic phase, yeast proteins only produce an exothermic phase.
DOI: https://doi.org/10.7554/eLife.41331.013

The following figure supplement is available for figure 4:

**Figure supplement 1.** Size exclusion peaks of RanBP1, RanGTP and CRM1 from yeast or animals mixed at 2:5:1 molar ratio and SDS-PAGE analysis of different peaks.
DOI: https://doi.org/10.7554/eLife.41331.014

possibly those cells did not have a detectable level of nuclear RanBP1 (*Figure 5B* white arrows). To generate an appropriate negative control for this experiment, we mutated C-terminus of Ran (A181W, P184H and L209H, named Ran$^{Cmut}$, single mutation is insufficient to disrupt its tight binding to RanBP1), so that it is still active (able to promote CRM1-NES binding) but no longer binds to RanBP1, therefore unable to form RanBP1 tetramer (*Figure 5C*). When the two Ran$^{Cmut}$ fusion constructs were co-transfected, no YFP signal was observed (*Figure 5B*), suggesting that the weak signals that we observed earlier using Ran$^{WT}$ are true signals.

The weak signals were probably due to low level of RanBP1 in the nucleus (*Richards et al., 1996*) and competition by endogenous RanGTP. In order to enhance the signals that we observed, we further transfected NLS$^{SV40}$-fused hRanBP1 or hNESmut (which is the human version of NESmut) to artificially increase the nuclear pool of RanBP1 along with the two Ran$^{WT}$ fusion plasmids. In contrast to NLS-hRanBP1 which significantly promoted nuclear YFP formation, NLS-hNESmut inhibited the weak YFP signals observed in the absence of NLS-hNESmut (possibly inhibiting tetramer formation by sequestering Ran fusions) (*Figure 5B*). This further demonstrates that NES is the only interface between mRanBP1 and CRM1, without which the formation of tetramer is inhibited. Consistent with the fact that RanGTP is almost absent in cytoplasm (*Izaurralde et al., 1997*), strong cytoplasmic YFP signals were not observed during image collection, validating that the complex forms only in excess of RanGTP. As an alternative approach, we measured the Fluorescence Resonance Energy Transfer (FRET) efficiency between CFP-Ran and Alexa Fluor 546 immuno-labelled Myc-Ran, while co-transfecting either RanBP1$^{\Delta linker}$-NLS (forms tetramer; linker refers to residues between RBD and NES; linker was deleted to reduce distances between two Rans and enhance FRET efficiency) or RanBP1$^{\Delta NES}$-NLS (does not form tetramer). In agreement with the BiFC experiment, the nuclear FRET efficiency is significantly (p<0.0001) higher when transfected with RanBP1$^{\Delta linker}$-NLS (*Figure 5— figure supplement 2*). In summary, we detected the RanBP1 tetrameric complex in (and only in) the nucleus of human cells.

## Reasons for not forming trimeric RanBP1-RanGTP-CRM1 complex in animals

The key nuclear export and cargo dissociation mechanistic change from yeast to animals is the loss of binding between RanBP1-RanGTP and CRM1. We then asked why animal RanBP1-RanGTP did not bind to CRM1 as the yeast proteins. Through a pull down assay to test inter-species CRM1-RanGTP-RanBP1 complex formation, we found that in contrast to yRanBP1-hRanGTP-yCRM1 complex, neither yRanBP1-hRanGTP-hCRM1 nor ΔC-hRanGTP-yCRM1 complex was formed (*Figure 6A*). Therefore, neither hCRM1 nor ΔC is compatible for trimeric complex formation. We previously showed that yRanBP1 was unexpectedly localized in the nucleus of HeLa cells (*Figure 1F*). This could be explained by *Figure 6A*: yRanBP1 does not form trimeric or tetrameric complex with human CRM1/RanGTP, thus is not exported to the cytoplasm of HeLa cells.

It should be noted that human and yeast Ran proteins are not critical for trimeric complex formation. In this study, we discovered a small difference between human and yeast Ran, that is hRan-yRanBP1-yCRM1 complex has lower affinity than yRan-yRanBP1-yCRM1 (*Figure 4F*; refer also to *Maurer et al., 2001*). This reduction in affinity agrees with pull down that hRan-yRanBP1-yCRM1 but not yRan-yRanBP1-yCRM1 could be dissembled by NES (which is the reason for using yRan but not hRan for yeast complex in cargo dissociation experiments in *Figure 2*). Though the affinity of Ran-

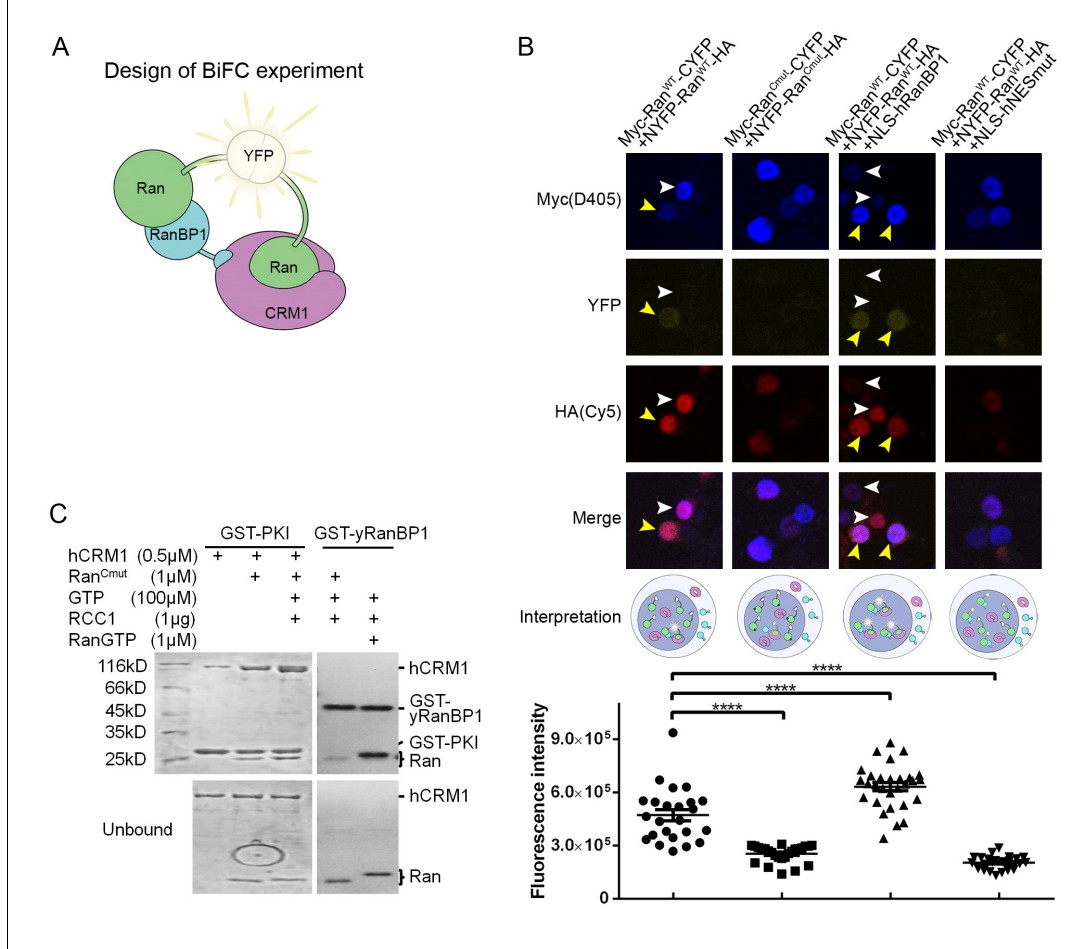

**Figure 5.** RanBP1 tetrameric complex exists in the cell nucleus. (**A**) Design of BiFC experiment. When the RanBP1 tetramer is formed, a YFP signal could be detected. This assay may not discriminate which Ran binds to RanBP1 (or CRM1) in the tetramer. (**B**) Representative immunofluorescence (HA and Myc) and fluorescence (YFP) images on cells co-transfected with fusion plasmids (labelled on top). NLS-hRanBP1 or NLS-hNESmut (hNESmut: human RanBP1 with NES mutation) was used to increase nuclear level of RanBP1 or its mutant. Nucleus was not stained so that the two transfected proteins could be stained without contaminating the YFP channel. The boundary of nucleus could be estimated by Ran fusions, which is mainly localized in the nucleus (see *Figure 5—figure supplement 3*). Results interpretation (middle panel) explains why the yellow fluorescence is observed or not. Bottom panel shows the nuclear YFP intensities of at least 20 cells (only florescent cells transfected with two Ran fusions) in the corresponding samples under the same level of illumination. **** denotes p<0.0001. (**C**) Ran$^{Cmut}$ is able to bind to CRM1 and NES (left panel) but not able to bind yRanBP1 (right panel).

DOI: https://doi.org/10.7554/eLife.41331.015

The following figure supplements are available for figure 5:

**Figure supplement 1.** Pull down of 293 T cells expressed Ran fusion proteins using GST-mRanBP1 or GST, in the presence or absence of Ran knock down.
DOI: https://doi.org/10.7554/eLife.41331.016

**Figure supplement 2.** Acceptor photobleach-FRET and statistical analysis of FRET efficiencies.
DOI: https://doi.org/10.7554/eLife.41331.017

**Figure supplement 3.** Immunofluorescence (HA and Myc) on cells co-transfected with two fusion plasmids.
DOI: https://doi.org/10.7554/eLife.41331.018

yRanBP1-yCRM1 is significantly reduced with hRan, we showed that hRan-yRanBP1-yCRM1 complex formed readily by pull down (*Figure 6A*), and this complex is competent in cellular nuclear export (will be presented in a later section).

The above results suggested that essential residues for trimeric complex formation in yeast CRM1/RanBP1 were not conserved in animals. When crystal structures of hRanBP1-hRanGTP (pdb:1K5G) and hCRM1 (pdb:3GB8) were aligned onto yCRM1-hRan-yRanBP1 (pdb:4HAT), several

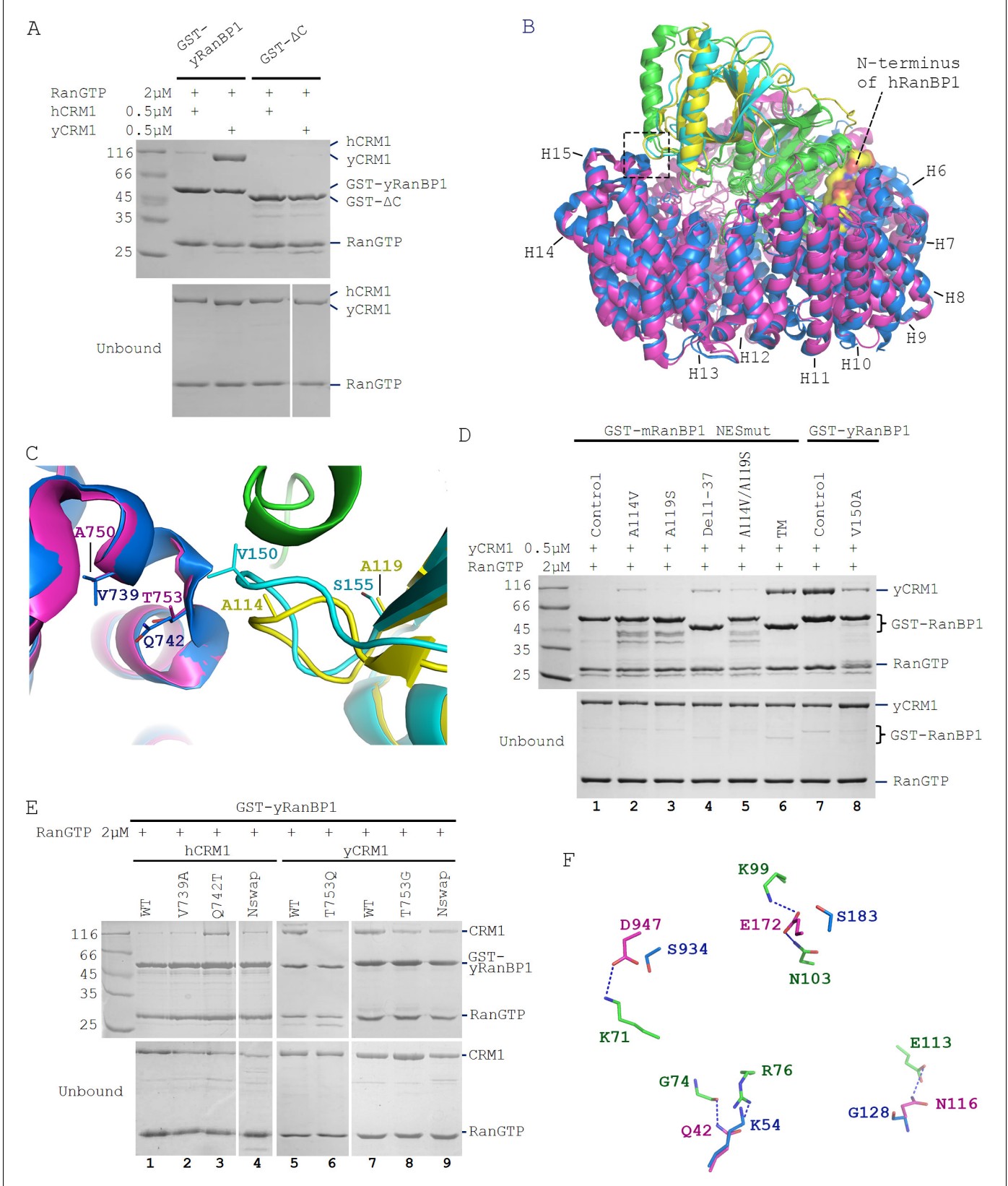

**Figure 6.** Sequence deviation of RanBP1 and CRM1 from their yeast orthologs prevents binding between RanBP1-RanGTP and CRM1 in animals. (A) Cross-species pull down of CRM1 using immobilized GST-yRanBP1 or GST-ΔC in the presence of RanGTP. (B) Superimposition of hRanBP1-hRan (1K5G,

*Figure 6 continued on next page*

Figure 6 continued

yellow and green) and hCRM1 (3GB8, blue) onto yRanBP1(cyan)-hRan(green)-yCRM1(magenta) crystal structure (4HAT). N-terminus of hRanBP1 (surface representation) is located between intimately packed RanGTP and CRM1 and thus possibly hinders their binding in animals. Heat repeats 6–15 (H6–H15) of CRM1 are labelled. (C) Zoom in of boxed region in 6B shows RanBP1-CRM1 interface and adjacent residues (shown as sticks) that are different between species. (D) GST-RanBP1 and its mutants pull down of yCRM1 and RanGTP. TM contains mutations A114V, A119S and Δ1–37, in addition to NES mutation. A114V and N-terminal deletion collectively rescue binding to yCRM1-RanGTP. (E) GST-yRanBP1 pull down of human and yeast CRM1 mutants in the presence of RanGTP. Both RanBP1 and Ran interacting regions of CRM1 are important for trimeric complex formation. (F) In the yRanBP1-hRan-yCRM1 crystal structure, yCRM1 and hRanGTP residues that form hydrogen bonds (dash lines) are displayed as magenta sticks. The structurally non-conserved hCRM1 residues are displayed as blue sticks. To improve clarity, main chain atoms (except glycine) and conserved residues are omitted.

DOI: https://doi.org/10.7554/eLife.41331.019

The following figure supplements are available for figure 6:

**Figure supplement 1.** Multiple sequence alignment of fungi RanBP1 proteins to show that S.
DOI: https://doi.org/10.7554/eLife.41331.020

**Figure supplement 2.** Residues that form hydrogen bonds (blue dash) between hRanGTP (green) and yCRM1 (magenta) in 4HAT structure.
DOI: https://doi.org/10.7554/eLife.41331.021

**Figure supplement 3.** Sequence alignment of human and yeast Ran.
DOI: https://doi.org/10.7554/eLife.41331.022

**Figure supplement 4.** Sequence alignment of human and yeast CRM1.
DOI: https://doi.org/10.7554/eLife.41331.023

differences between RanBP1 and CRM1 of the two species were observed (*Figure 6B,C*). First, unlike yRanBP1 (17), hRanBP1 residues preceding its RBD domain are ordered and inserted between the closely-packed RanGTP and CRM1 (*Figure 6B*), possibly hindering trimeric complex formation by steric clash. Second, V150 that is in close proximity with yCRM1 in yRanBP1 corresponds to A114 in hRanBP1 (*Figure 6C*). V150 is conserved to be valine or isoleucine in fungi (*Figure 6—figure supplement 1*), but not conserved in hRanBP2 and RanBP1 from human, mouse, zebrafish and frog (*Figure 1—figure supplement 4*). The residues discussed above were mutated in NESmut and yRanBP1 to test whether they impact binding towards yCRM1 in the presence of RanGTP. Pull down results showed that either A114V or deletion of N terminus (Del 1–37) on NESmut could partially rescue binding to yCRM1 (*Figure 6D*, lane 2 and 4). In agreement, V150A mutation in yRanBP1 partially inhibited binding of yCRM1 (*Figure 6D*, lane 7 and 8). In contrast, mutation of more distant residue (A119 in NESmut mutated to yeast-equivalent S) did not significantly affect binding. Triple mutant (A114V, A119S and Del1-37, labelled as TM) rescued yCRM1 binding to a comparable level as yRanBP1 (*Figure 6D*, lane 6,7). Taken together, both the N-terminus and CRM1 interacting region of animal RanBP1 are incompatible for trimeric complex formation.

Similarly, several regions of CRM1 were mutated to identify critical sequence changes that abolished the formation of trimeric complex in animals. In the yRanBP1 binding interface, side chain of the strictly conserved yCRM1 T753 is in close contact with yRanBP1 V150 (*Figure 6C*). The corresponding residue in human is Q742, being strictly conserved in animals. Pull down results showed that Q742T mutation partially rescued binding between hCRM1 and yRanBP1-RanGTP (*Figure 6E*, lane 3). Similarly, T753Q or T753G mutation in yCRM1 also reduced its binding to yRanBP1-RanGTP (*Figure 6E*, lane 5–8). V739A mutation in hCRM1 (V739 equivalent residues in yCRM1 is A), however, did not rescue binding (*Figure 6E*, lane 2).

Besides the yRanBP1 interaction surface discussed above, yRanBP1 binds to yCRM1 indirectly through RanGTP. Human RanGTP forms 26 hydrogen bonds with yCRM1 in yRanBP1-hRan-yCRM1 structure (*Figure 6—figure supplement 2*), and those residues that form hydrogen bonds are strictly conserved in yeast Ran (*Figure 6—figure supplement 3*). However, six hydrogen bonds are predicted to be lost if yCRM1 is replaced with hCRM1 (*Figure 6F*, *Figure 6—figure supplement 4*). Five hydrogen bonds are contributed by N-terminal residues (E172, Q42, N116) in yCRM1, and one by a C terminal residue (D947) (*Figure 6F*, *Figure 6—figure supplement 2*). Instead of making single mutations individually, the N-terminal 200 residues of hCRM1 were replaced with yeast equivalent residues 1–188, and the chimera protein (hCRM1$^{N\_swap}$) was able to weakly bind to yRanBP1-RanGTP (*Figure 6E*, lane 4). Similarly, when the N-terminal 1–188 region of yCRM1 was replaced by human equivalent N-terminal region 1–200 (yCRM1$^{N\_swap}$), binding to yRanBP1-RanGTP was

significantly reduced (*Figure 6E*, lane 9). Pull down assay did not produce a significant difference by further mutating D947 (data not shown). Therefore, loss of binding between hCRM1 and yRanBP1-RanGTP is due to sequence divergence of hCRM1 from yCRM1, including the RanBP1 contact surface and Ran contact surfaces.

## Identified mutations play important roles in RanBP1 localization and nuclear export of cargo

Previously, we showed that yeast RanBP1 is localized in the nucleus in HeLa cells. When yCRM1 was co-transfected with yRanBP1, yRanBP1 was significantly (p<0.0001) relocalized to the cytoplasm (*Figure 7A,B*), suggesting that yCRM1 but not endogenous hCRM1 formed a nuclear export complex with yRanBP1. In contrast, co-transfected yCRM1$^{T753Q}$, the mutant that displayed reduced binding to yRanBP1-RanGTP by pull down, barely promoted nuclear export of yRanBP1 (p<0.0001). Similarly, the predominantly nucleus-localized tipple mutant of NESmut (NESmut) was significantly (p<0.0001) exported to the cytoplasm when co-transfected with yCRM1, but not yCRM1$^{T753Q}$ (*Figure 7C,D*). These cellular studies fully recapitulated what was observed by pull down (*Figure 6*), supporting the molecular mechanism for loss of trimeric complex formation in animals.

Endogenous human RanBP1 knock-down and further rescue with mCherry-tagged yRanBP1, yRanBP1$^{V150A}$, mRanBP1, ΔC or NESmut did not change the localization of endogenous CRM1, the nuclear export cargo NFκB and the transfected nuclear import cargo GFP-NLS (*Figure 7—figure supplement 1*). This is probably due to the presence of functionally redundant RanBP2 in mammals (*Villa Braslavsky et al., 2000*), though RanBP1 is reported to be an essential gene in yeast (*Petersen et al., 2000*). Cargo's proper localization also suggests that the nucleus concentration of mis-localized RanBP1 is probably too low to disrupt the cellular RanGTP gradient (which is critical for nuclear transport) (*Izaurralde et al., 1997*). On the other hand, cytoplasmic localization of endogenous RanBP1 and NFκB was inhibited by siCRM1, which could be rescued by the expression of either hCRM1 and yCRM1$^{T753Q}$, but not yCRM1 and hCRM1$^{Q742T}$ (*Figure 7E–G*). Though hCRM1$^{Q742T}$ is much more like hCRM1 than yCRM1$^{T753Q}$ in sequence, function-wise the opposite is true, suggesting that Q742 (T753 in yeast) is an important function-defining residue. Moreover, hCRM1$^{Q742T}$ displayed strong nuclear rim staining in all acquired images (*Figure 7E*), which might be due to the formation of weakly GAP-resistant CRM1-RanGTP-RanBP2 trimeric complex. Negative control using transfected mCherry-NLS displayed no localization changes under siCRM1 and different rescue conditions (*Figure 7H*).

## Higher efficiency of nuclear transport in human cells

We previously showed that when CRM1 is limited, yeast proteins (that form tight trimeric complex) are less efficient than human proteins in nuclear export of cargoes. However, one might argue that yeast proteins are innately less efficient when used in human cells. We therefore performed a similar experiment using all yeast proteins, that is WT proteins that form trimer, and mutant yeast proteins that do not (*Figure 8A*). The mutations include yCRM1$^{T753Q}$ and yRanBP1$^{V150A}$, which inhibits trimer formation (*Figure 6D,E*), and fusion of NES$^{mRanBP1}$ to C-terminus of yRanBP1$^{V150A}$ to prevent nuclear accumulation of yRanBP1. Statistical analysis showed a faster nuclear export rate for mutant than WT proteins (*Figure 8B*), reconfirming that the nuclear export rate is higher when not forming tight trimer.

By doubling the energy expenditure, animal cells may be able to do 'more work' and could export against a higher nucleo-cytoplasmic concentration difference. To test this hypothesis, we analyzed the subcellular localization of transfected GFP-NLS and mCherry-NES in human (HeLa) and yeast cells. Astonishingly, while mCherry-NES showed predominant cytoplasmic localization (22% nuclear) in human cells, the same protein in yeast cells showed much higher nucleus localization level (43% nuclear) (*Figure 8C,D*), though mCherry-NES bound to yeast and human CRM1 transportors with similar strength (*Figure 8—figure supplement 1*). Similarly, GFP-NLS, which displayed similar binding strength to human and yeast transportors (*Figure 8—figure supplement 1*), was much less nucleus localized in yeast cells (59% nuclear) compared to in human cells (82% nuclear) (*Figure 8C, D*). In addition, mCherry-NES were expressed at very low level in yeast cells compared to mammalian cells (*Figure 8—figure supplement 2A*), while the Ran and CRM1 concentrations in both cells are comparable (*Figure 8—figure supplement 2B,C*), suggesting that the lower export efficiency in

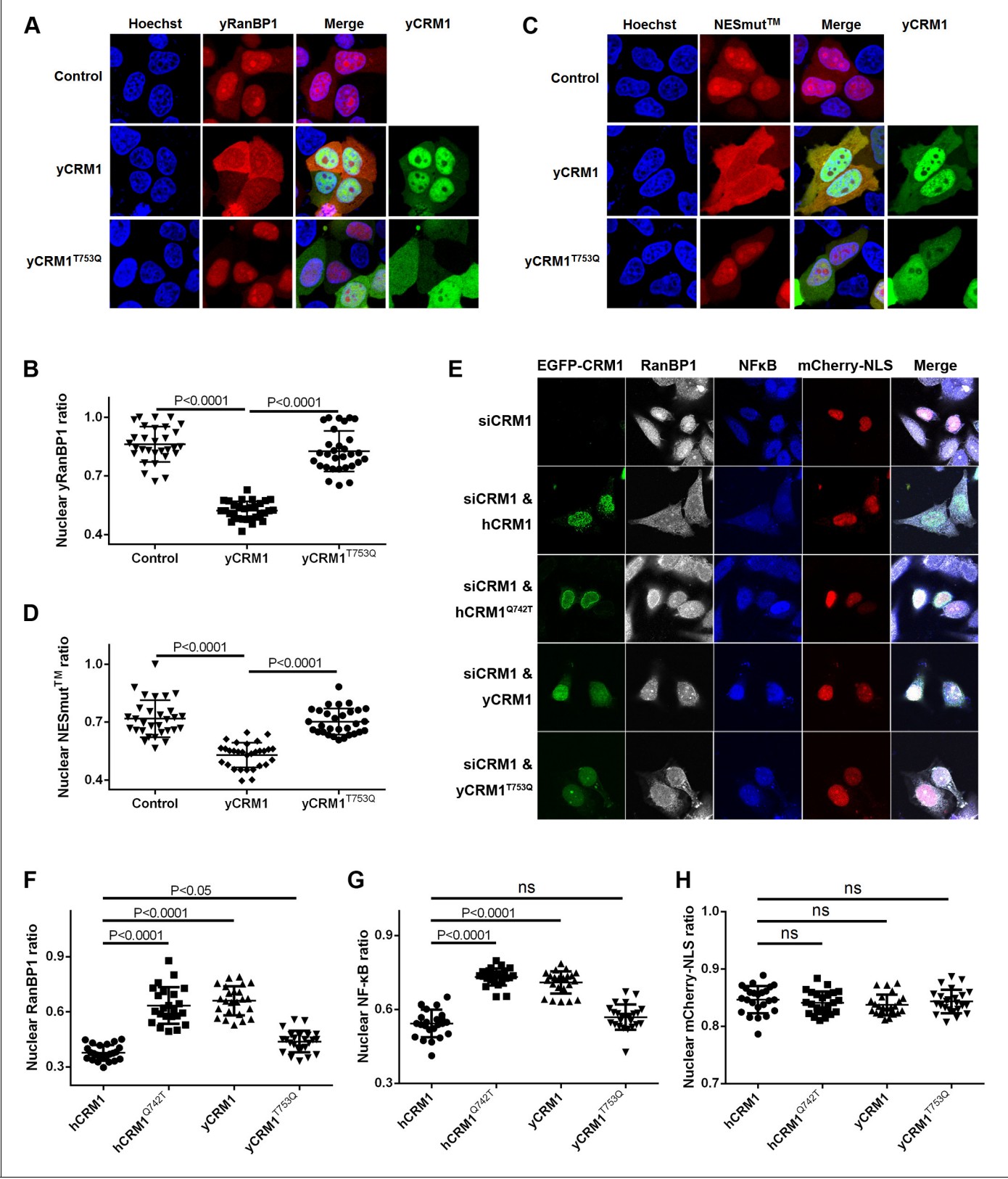

**Figure 7.** Residues that play important roles in RanBP1 localization and nuclear export of cargo. (**A**) mCherry tagged yRanBP1 was transfected alone or co-transfected with EGFP-yCRM1 or EGFP-yCRM1^T753Q mutant, and visualized under microscope. (**B**) Quantification and statistical analysis of yRanBP1 localization in 7A. Ratio of nuclear yRanBP1 for each cell is calculated as its nuclear intensity divided by total cellular intensity. Error bars represent

*Figure 7 continued on next page*

*Figure 7 continued*

standard deviation of each set of data containing measurements from at least 30 cells. (C) mCherry tagged NESmut was transfected alone or co-transfected with EGFP-yCRM1 or EGFP-yCRM1$^{T753Q}$ mutant, and visualized under microscope. (D) Quantification and statistical analysis of NESmut localization in 7C. (E) Subcellular localization of endogenous RanBP1, NFκB and transfected mCherry-NLS under the treatment of siCRM1 and transfection of different EGFP-CRM1 constructs. mCherry-NLS is constructed as mCherry-NES$^{PKI}$-MBP-NLS$^{SV40}$ in pmCherry-C1 plasmid. (F–H) Quantification and statistical analysis of nuclear ratio of RanBP1, NFκB and mCherry-NLS in 7E.
DOI: https://doi.org/10.7554/eLife.41331.024

The following figure supplement is available for figure 7:

**Figure supplement 1.** siRanBP1 does not change the subcellular localization of endogenous CRM1, NFκB and transfected GFP-NLS.
DOI: https://doi.org/10.7554/eLife.41331.025

yeast should not be due to saturation of export capacity. In summary, these results advocate the notion that nuclear transport efficiency is higher in human cells than in yeast cells.

## Discussion

### Mechanistic difference between fungi and animal RanBP1 nuclear export

Nuclear export of RanBP1 is essential for all eukaryotic cells since excess of RanBP1 in the nucleus sequesters nuclear RanGTP, inhibits nuclear transport and is toxic to cells (*Izaurralde et al., 1997*; *Richards et al., 1996*). In both yeast and human, RanBP1 is exported by nuclear export factor CRM1 (19). Though yeast and mammalian proteins are used in this work, the mechanisms revealed in this study should be applicable to fungi and animals. Due to the lack of NES, fungi RanBP1 should form trimeric complex through its RBD interacting with both RanGTP and CRM1. This explains the observation that intact RBD is required for its nuclear export (*Künzler et al., 2000*). Obviously, NES is not needed for its nuclear export because fungi RBD is sufficient for its nuclear export. In animals, NES but not RBD is required for its export because animal RBD-RanGTP does not bind to CRM1 at sufficient high affinity to be exported. We show that this is due to degeneration of the interface residues essential for the formation of trimeric complex.

Since nucleus is enriched with RanGTP, a tetrameric animal export complex of (RanBP1-RanGTP)-CRM1-RanGTP would form in animals, whereby RanBP1-RanGTP binds to the NES groove of CRM1 (through the NES of RanBP1) as an NES cargo. It should be noted that animal tetrameric complex probably exists only in the nucleus, whereas the fungi trimeric complex could exist both in nucleus and cytoplasm. In the nucleus, animal RanBP1 cargo dissociation activity is inhibited with excessive RanGTP, and it would not dissociate any CRM1 cargo or its own NES before reaching the cytoplasm. When animal RanBP1 tetrameric complex enters the cytoplasm, where RanBP1/RanGAP is present at high concentration and RanGTP at low concentration, the complex would probably be quickly dissembled (shutting off cytoplasmic YFP signals in the BiFC experiment).

### Mechanistic difference between fungi and animal RanBP1's cargo dissociation

In the cytoplasm, fungi and animal RanBP1 also use different mechanisms to displace CRM1 cargoes. Both fungi and animals use RBD domains for cargo dissociation, and we showed that NES of animal RanBP1 is dispensable for cargo dissociation. Fungi RanBP1 binds tightly to RanGTP-CRM1 and closes the NES groove to actively release bound NES. This ends up sequestering available CRM1, before RanGTP is hydrolyzed with the help of RanGAP. In animals, excessive RanBP1 (and RanBP2) in the cytoplasm sequesters RanGTP from the NES-CRM1-RanGTP complex, resulting in a transient low-affinity binary CRM1-NES complex, which dissembles automatically. However, we do not exclude the possibility that a transient complex of animal RanBP1-RanGTP-CRM1 forms like the fungi complex, which actively dissociates NES from CRM1, immediately followed by dissociation of RanBP1-RanGTP-CRM1. In fact, active displacement of cargo was observed previously with the yCRM1 mutant (P754D), which is defective in the stable yRanBP1-yRanGTP-yCRM1 ternary complex formation (*Koyama and Matsuura, 2010*). Consistent with the proposed cargo dissociation

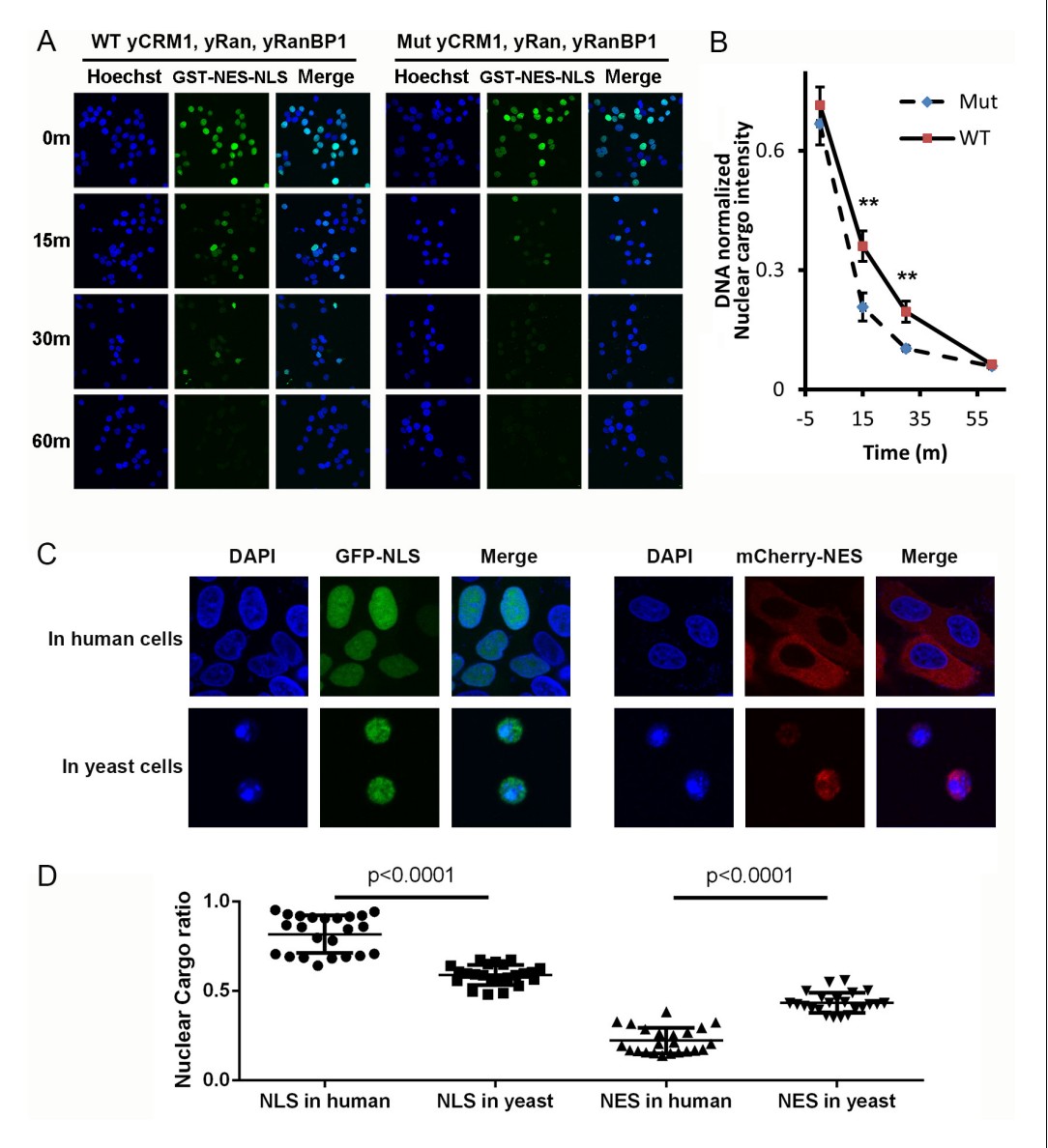

**Figure 8.** Mammalian nuclear transport system has higher efficiency. (**A**) Nuclear export of GST-NES-NLS in the presence of yCRM1/yRan/yRanBP1 or their mutants (yCRM1[T753Q], yRan, yRanBP1[V150A and fused with mRanBP1's NES]). Protocol used in this experiment is similar as in *Figure 3C*. (**B**) Quantification of nuclear cargo intensity normalized by DNA in *Figure 7E*. Shown also includes unpaired student t-test between WT and mutant samples at each time point. Error bars represent standard error of measurements for each set of data containing measurements from at least 23 cells. Only 15 m and 30 m samples display statistical significances (** denotes p<0.01). (**C**) Representative images of GFP-NLS and mCherry-NES localization in human (HeLa) and yeast cells (W303.1a). pRS416 (for yeast transfection), pEGFP-C1 and pmCherry-C1 (for human transfection) plasmids were used to express fluorescent tagged NES or NLS proteins. mCherry-NES is constructed as mCherry-NES[PKI]-MBP-NLS[SV40]. GFP-NLS is constructed as GFP-NES[PKI]-MBP-NLS[BPSV40]. (**D**) Quantification and statistical analysis of nuclear cargo ratio in 8C.
DOI: https://doi.org/10.7554/eLife.41331.026

The following figure supplements are available for figure 8:

**Figure supplement 1.** The NLS/NES used displayed similar affinities to yeast or human transport receptors.
DOI: https://doi.org/10.7554/eLife.41331.027
**Figure supplement 2.** Yeast mCherry-NES-NLS were expressed at very low level compared to human cells.
DOI: https://doi.org/10.7554/eLife.41331.028

mechanisms, we show that in the presence of RanGAP and RanBP1, yeast but not human CRM1 partially inhibits RanGTP hydrolysis.

Besides dissociating cargo from CRM1, RanBP1 is essential for the disassembly of tightly-bound Ran-karyopherin complexes. In fact, the Ran-sequestering mechanism seems to be more prevalent among RanBP1 mediated RanGTP dissociation from different karyopherins. Only human and yeast Importin β1 (*Lounsbury and Macara, 1997*; *Deane et al., 1997*; *Floer et al., 1997*; *Bischoff et al., 1995a*), and yeast CRM1 (*Künzler et al., 2000*; *Maurer et al., 2001*), form very tight complexes with RanBP1-RanGTP, which are partially GAP resistant. For human importin β2 (*Lounsbury and Macara, 1997*), importin β3 (*Deane et al., 1997*), yeast importin 4 (*Schlenstedt et al., 1997*), human importin 7 (*Görlich et al., 1997*), human importin 8 (*Görlich et al., 1997*), human Exportin-1 (*Paraskeva et al., 1999*; *Askjaer et al., 1999*), human Exportin-2 (*Kutay et al., 1997*; *Bischoff et al., 1995a*) and human exportin-t (*Kutay et al., 1998*), their affinities with RanGTP-RanBP1 are relatively weak and are effectively dissembled in the presence of RanGAP. In the case of human and yeast KPNB1, NLS is required for efficient release of Ran-RanBP1 for GAP hydrolysis (*Bischoff and Görlich, 1997*; *Floer et al., 1997*). For yCRM1, there is no corresponding RanBP1-RanGTP release factor reported thus far.

## Biological significances of forming trimeric or tetrameric complex

Apparently, one round nuclear export of fungi RanBP1 costs energy of hydrolysing one GTP, while one round nuclear export of animal RanBP1 costs energy of two GTPs. Since RanBP1 is a constantly shuttling protein, animals may spend significantly more energy than fungi in the long term. We believe that at the same time of suffering this disadvantage in energy consumption, forming tetrameric complex may also confer at least two advantages for animals. First, as mentioned above, forming tetrameric complex does not inhibit GAP mediated hydrolysis, which means faster recycling rate of nuclear export machinery. This possibly further implies less congestion of nuclear pore and faster rate of nuclear transport, or translated into a higher nucleo-cytoplasmic concentration difference of cargoes. Indeed, we found that both nuclear import and export of cargoes were substantially more efficient in human cells than in yeast cells (*Figure 8C*). Though we showed that this is not due to differences in binding strength to transport receptors nor cargo expression levels in two types of cells, our results do not eliminate other possibilities such as a leakier NPC in yeast; further experiments are needed to fully understand how much the yeast RanBP1 system (which partially inhibits GAP hydrolysis) contributes to the observed lower concentration gradient of cargoes. Second, the interface region and several other regions in fungi have to be conserved in order to maintain trimeric complex formation and export nuclear RanBP1. In animals, the existence of specialized NES in RanBP1 has eliminated that requirement, allowing multiple regions in both RanBP1 and CRM1 to diverge and participate in other yet-to-be discovered cellular functions. Compared with fungi, animals are readily mobile and more complex in cellular/tissue organization. Whether the appearance of RanBP1 NES in animals is involved in higher order cellular organization, tissue development or independent mobility, and whether the tetrameric complex plays roles in other processes warrant further studies.

Further, the different mechanisms observed in this study could aid in development of inhibitors against fungi by targeting the critical binding interfaces discussed above to inhibit the formation of trimeric RanBP1 nuclear export complex. Anti-fungal medicines developed by this strategy are potentially safe for human beings, since our RanBP1 nuclear export system does not rely on the formation of the trimeric complex.

## Materials and methods

**Key resources table**

| Reagent type (species) or resource | Designation | Source or reference | Catalogue no. | Additional information |
|---|---|---|---|---|
| Genetic reagent (H. sapiens) | pcDNA 3.1(+) myc-Ran$^{WT}$-CYFP | this study | | Used in BiFC |

*Continued on next page*

Continued

| Reagent type (species) or resource | Designation | Source or reference | Catalogue no. | Additional information |
|---|---|---|---|---|
| Genetic reagent (H. sapiens) | pcDNA 3.1(+) NYFP-Ran$^{WT}$-HA | this study | | Used in BiFC |
| Genetic reagent (H. sapiens) | pcDNA 3.1(+) myc-Ran$^{Cmut}$-CYFP | this study | | Used in BiFC |
| Genetic reagent (H. sapiens) | pcDNA 3.1(+) NYFP-Ran$^{Cmut}$-HA | this study | | Used in BiFC |
| Genetic reagent (H. sapiens) | pcDNA 3.1(+) CFP-RanWT | this study | | Used in FRET |
| Genetic reagent (H. sapiens) | pcDNA 3.1(+) myc-RanWT | this study | | Used in FRET |
| Genetic reagent (H. sapiens) | pcDNA 3.1(+) hRanBP1$^{\Delta linker}$-NLS | this study | | Used in FRET |
| Genetic reagent (H. sapiens) | pcDNA 3.1(+) hRanBP1$^{\Delta NES}$-NLS | this study | | Used in FRET |
| Genetic reagent (M. musculus) | pcDNA 3.1(+) NLS-mRanBP1 | this study | | Used in BiFC |
| Genetic reagent (M. musculus) | pcDNA 3.1(+) NLS-NESmut | this study | | Used in BiFC |
| Antibody | Rabbit anti-HA | Cell Signalling | 3724 | |
| Antibody | Mouse anti-Myc | ProteinTech | 60003–2-Ig | |
| Antibody | Alexa 647 labelled Anti-Rabbit IgG | ThermoFisher | A-21244 | |
| Antibody | Dylight 405 labelled Anti-mouse IgG | Beyontime | A0609 | |
| Antibody | Anti-GST | Santa Cruz | sc-138 | |
| Antibody | Anti-RanBP1 | ProteinTech | 27804–1-AP | |
| Antibody | Anti-CRM1 | absin | 115104 | |
| Antibody | Anti-NFκB | SAB | 48676–1 | |
| Chemical compound | Digitonin | Abcam | ab141501 | |

## Cloning, protein expression and purification

The human, mouse or yeast RanBP1 (or their mutants) was cloned into a pGEX-4T1 based expression vector incorporating a TEV-cleavable N-terminal GST-tag fusion. The plasmid was transformed into *Escherichia coli* BL-21 (DE3) and grown in LB Broth medium. Expression of protein was induced by the addition of 0.5 mM isopropyl β-D-1-thiogalactopyranoside (IPTG), and the culture was grown overnight at 18 ˚C. Cells were harvested and sonicated in lysis buffer (50 mM Tris pH 8.0, 200 mM NaCl, 10% glycerol, 2 mM DTT, 1 mM EDTA and 1 mM PMSF). RanBP1 was purified on a GST column and eluted after TEV cleavage in a buffer containing 20 mM Tris pH 8.0, 200 mM NaCl, 10% glycerol, 1 mM EDTA and 2 mM DTT. This was followed by a Superdex 200 increase gel filtration column on the ÄktaPure (GE Healthcare) using the gel filtration buffer (20 mM Tris pH 8.0, 200 mM NaCl, 10% glycerol, 2 mM DTT). Eluted proteins were frozen at −80˚C at 5–10 mg/ml. Alternatively, GST-RanBP1 was eluted with 20 mM Tris pH 8.0, 200 mM NaCl, 1 mM EDTA, 2 mM DTT, 10 mM reduced glutathione, and purified by Superdex 200 increase column. Both mouse and human RanBP1 have been used in this study and they are highly similar (95% identity, *Figure 1—figure supplement 4*). ΔC construct contains deletion of C terminal residues' PGKNDNAEKVAEKLEALSVREA REEAEEKSEEKQ'. His-tagged proteins (human and yeast CRM1 and related mutants) were expressed in *E. coli* grown in TB Broth medium. The proteins were induced in the presence of 0.5 mM IPTG overnight at 25 ˚C, and purified by Nickel beads. 6 × His tagged proteins were eluted with 300 mM Imidazole pH 7.5, 300 mM NaCl, 10% glycerol, and 2 mM BME.

## Pull down assay

All proteins used were purified by S200 prior to pull down. To assess different interactions, GST-tagged proteins were immobilized on GSH beads. For GST-RanBP1, a wash step was performed immediately after immobilization to remove unbound GST-RanBP1. Soluble proteins at indicated concentrations were incubated with the immobilized proteins in a total volume of 1 ml for two hours at 4˚C. After two washing steps, bound proteins were separated by SDS/PAGE and visualized by Coomassie Blue staining. Each experiment was repeated at least twice and checked for consistency. The pull down buffer contains 20 mM Tris pH 8.0, 200 mM NaCl, 10% glycerol, 5 mM $MgCl_2$, 0.005% Triton-X100, and 5 mM DTT if not specified. $Ran^{WT}$ contains about 95% RanGDP and is used as RanGDP (*Zhang et al., 2018*). Human $Ran^{L182A}$ and yeast $Ran^{L184A}$ used in this paper are C-terminus destabilized, therefore loaded with more than 80% of GTP, and are used as RanGTP (*Zhang et al., 2018*).

## Isothermal Titration Calorimetry (ITC)

ITC experiments were conducted at 20˚C using ITC200 (Microcal) in a buffer containing 20 mM Tris pH 8.0, 200 mM NaCl and 5 mM $MgCl_2$. Q69L and L182A double mutant of Ran were used in this assay because this Ran mutant is 100% GTP bound (*Zhang et al., 2018*). For animal complex, 250 µM Ran was titrated into the sample cell containing 20 µM hCRM1 and 20 µM mRanBP1. For yeast complex, 150 µM Ran was titrated into the sample cell containing 15 µM yCRM1 and 15 µM yRanBP1. Each experiment was repeated at least twice. Data were processed by NITPIC (*Scheuermann and Brautigam, 2015*) and fitted by SEDPHAT (*Houtman et al., 2007*).

## Subcellular localization imaging

pEGFP-C1 and pmCherry-C1 plasmids were used to express EGFP or mCherry-tagged CRM1 or RanBP1 constructs. HeLa and 293 T cells were obtained from the Cell Bank of Chinese Academic of Sciences (Shanghai, China). HeLa cells were maintained in Dulbecco's modified Eagles medium (Hyclone) supplemented with 10% (vol/vol) fetal bovine serum (Biological Industries). For experiments without siRNA, cells were transfected with TurboFect transfection reagent (ThermoFisher) and fixed after 24 hr of transfection. For siRNA work, cells were first transfected with siRanBP1 (GGGCAAAACUGUUCCGAUUUG) or siCRM1 (CUCAGAAUAUGAAUACGAATT) using Lipo2000 (ThermoFisher) (*Fan et al., 2011*). After 24 hr, cells were transfected with different plasmids in the presence of polyethylenimine (PEI) transfection reagent. Cells were visualized 48 hr after the second transfection. Antibodies against RanBP1 (ProteinTech, 1:400), CRM1 (absin, 1:400), NFκB (SAB, 1:250), HA (CST, 1:1000), and Myc (ProteinTech, 1:1000) were used. Images were acquired by Olympus FV-1000 confocal microscope, and were analyzed using NIH ImageJ and Graphpad software.

## Bimolecular fluorescence complementation

NYFP-$Ran^{WT}$-HA, Myc-$Ran^{WT}$-CYFP, NYFP-$Ran^{Cmut}$-HA and Myc-$Ran^{Cmut}$-CYFP were cloned into pCDNA3.1(+) expression vectors. 293 T cells were seeded in 24-well plates containing circular coverslips slides in Dulbecco's modified Eagles medium (Hyclone) supplemented with 10% (vol/vol) fetal bovine serum (Biological Industries), 100 U/mL penicillin and 100 µg/mL streptomycin in a 5% $CO^2$ atmosphere at 37˚C. Twenty-four hours later, cells were transfected with plasmids in 1 mg/ml PEI (Polyethylenimine, Polysciences) transfection reagent. After another 24 hr, cells were treated with 25 µM biliverdin to boost the YFP signals. Two hours later, cells were fixed and incubated with the primary antibody anti-HA (CST, rabbit) and anti-Myc (ProteinTech, mouse), then with secondary antibody anti-mouse (Beyotime, Dylight 405) and anti-rabbit (ThermoFisher, Cy5). Images were acquired by Olympus FV-1000 confocal microscope, and analyzed using NIH ImageJ software.

## GAP hydrolysis assay

Human (0.4 µM) or yeast wide type Ran (0.3 µM) protein was first mixed with hRanBP1 (1 µM) or yRanBP1 (1 µM), respectively. Then samples were briefly incubated with or without 1 µM CRM1 from respective species. Since Ran has a very slow intrinsic nucleotide exchange rate, 0.5 µM RCC1 and 100 µM of GTP was added to reload Ran with GTP after each cycle of GAP-mediated hydrolysis in all samples. The samples were then incubated with 0.3 µM yeast RanGAP for 0, 15, 45 and 75 min. Control samples were done with 1 µM CRM1, 1 µM GST-PKI and 0. 3 µM of Ran for both species.

The amount of free phosphate generated was measured using GTPase assay kit (Bioassay) and Multiskan FC microplate reader (ThermoFisher). Each reaction was repeated four times in parallel.

## In vitro nuclear export using semi-permeabilized cells

We performed in vitro nuclear export experiment with slight modifications (*Cassany and Gerace, 2009*). First, 1 µM GST-NES$^{PKI}$-NLS$^{IBB}$ (GST-NES-NLS), 1 µM KPNB1, 1 µM NTF2, 1 × energy regeneration system (*Cassany and Gerace, 2009*), 0.01% Triton-X100, and 2 µM of hRan were added to semi-permeabilized HeLa cells and incubated at room temperature for 60 mins to accumulate nuclear cargoes. The very low concentration of Triton-X prevents non-specific cytoplasmic binding, but does not permeate nuclear envelopes (*Figure 3—figure supplement 1*). After washing, the cells were incubated with either human or yeast proteins (0.1 µM CRM1, 1 µM Ran and 1 µM RanBP1), energy regeneration system, 0.01% Triton-X, for different time points at room temperature with gentle shaking. After reaction, the cells were washed, fixed and visualized by immunostaining with GST antibody (Santa Cruz, 1:400). Statistics were based on measurements from at least 30 cells for each sample, and statistical significance was calculated by one-way ANOVA test in Graphpad software.

## Conclusion

In summary, we have shown that animal RanBP1 binds to CRM1 only in the nucleus where there is excessive RanGTP, and forms a complex with CRM1 and two RanGTP proteins through its NES but not RBD. This complex is stable and is exported to the cytoplasm where it is dissembled by RBD containing proteins and RanGAP. In contrast to the CRM1-RanGTP sequestering cargo dissociation mechanism in fungi, animal RanBP1 dissociates cargo through stripping RanGTP from nuclear export complexes. The key difference between fungi and animals is that animal RanBP1-RanGTP does not bind to CRM1 as it does in fungi, due to degeneration of interface residues on both CRM1 and RanBP1, though either change is sufficient to reduce binding. The different mechanisms of RanBP1 nuclear export highlight how animals increase catalysis rate on the expense of more energy consumption. It is unexpected and interesting that while essential functions of orthologous proteins are conserved, profound differences exist in the underlining mechanisms.

## Acknowledgements

We thank Dr. Chong Chen, Dr. Yinglan Zhao and Dr. Bao Rui from Sichuan University for helpful discussions. Thanks Dr. Lei Qiu for proofreading this manuscript. This research is supported by Natural Science Foundation of China (NSFC) grants (#80502629 and #31671477).

## Additional information

### Funding

| Funder | Grant reference number | Author |
| --- | --- | --- |
| National Natural Science Foundation of China | 80502629 | Qingxiang Sun |
| National Natural Science Foundation of China | 31671477 | Da Jia |

The funders had no role in study design, data collection and interpretation, or the decision to submit the work for publication.

### Author contributions

Yuling Li, Jinhan Zhou, Sui Min, Resources, Data curation, Software, Formal analysis, Investigation, Methodology; Yang Zhang, Yuqing Zhang, Resources, Data curation; Qiao Zhou, Xiaofei Shen, Junhong Han, Resources, Supervision, Writing—review and editing; Da Jia, Resources, Supervision, Funding acquisition, Writing—review and editing; Qingxiang Sun, Conceptualization, Resources, Data curation, Software, Formal analysis, Supervision, Funding acquisition, Validation, Investigation, Visualization, Methodology, Writing—original draft, Project administration, Writing—review and editing

## Author ORCIDs

Qingxiang Sun (iD) https://orcid.org/0000-0002-9474-8882

## Decision letter and Author response

Decision letter https://doi.org/10.7554/eLife.41331.031
Author response https://doi.org/10.7554/eLife.41331.032

## Additional files

### Supplementary files

• Transparent reporting form
DOI: https://doi.org/10.7554/eLife.41331.029

### Data availability

All data generated or analysed during this study are included in the manuscript and supporting files.

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
