## [Decision Letter]

Thank you for sending your article entitled "Distinct RanBP1 nuclear export and cargo dissociation mechanisms between fungi and animals" for peer review at *eLife*. Your article has been evaluated by two peer reviewers and the evaluation has been overseen by a Reviewing Editor and Andrea Musacchio as the Senior Editor.

Summary:

This paper studies the mechanism of how RanBP1 triggers the cargo release from the nuclear export receptor Crm1. The authors find that – despite very high sequence conservation – animal RanBP1 functions differently than its yeast counterpart. Intriguingly, the animal protein seems to require two RanGTP molecules for complex dissociation, which would argue that nuclear protein export in animals requires twice as much energy as in yeast.

While the paper contains interesting observations for the field, there are several weaknesses that the authors will have to address before the paper can be considered for publication.

Essential revisions:

1) The paper largely describes the differences between the animal and yeast complexes but the physiological relevance of the findings remains unclear. The authors discuss some of the implications but it would be important to test at least some of these possibilities. Of note, by doubling the energy expenditure, animal cells should be able to do 'more work' and could export against a higher concentration difference. Is there any evidence that this is the case in animal cells?

2) On a similar note, can yeast cells lacking yCRM1 and yRanBP1 be complemented with the two animal counterparts (or vice versa)? Or does the triple mutant rescue the yeast deletion?

3) Why do both ΔNES and ΔC trigger cargo release, yet both these domains are required for efficient binding? What is the phenotype of ΔNES or ΔC mutants when expressed in tissue culture upon knockdown of the endogenous proteins?

4) It is not satisfying that the authors write, we 'believe' that the NES doesn't contribute to dissociation vial competition (subsection “Mouse RBD dissociates cargo through sequestering RanGTP”). That needs to be more vigorously tested. Similarly, they write that the C-terminus is 'required' for CRM1 cargo dissociation. It would be correct to say it is sufficient, but they don't show that it is required (as ΔC still dissociates in vitro). As mentioned above (point 1) it would be important to test whether the C-terminus is needed in vivo.

5) Why does the stable BIFC complex remain in the nucleus at steady state? If functional, it might be expected that the artificial Ran dimer might get stuck in the cytoplasmic after export? Is the BIF Ran complex functional and can it still shuttle? Also, the cells in Figure 5 (particularly the 3rd row) do not look very healthy and the results should be quantified.

6) The description of the in vitro nuclear export assay (in the Materials and methods) is not adequate and it remains unclear how these experiments were performed. Do they first perform nuclear import reactions, then a nuclear export reaction (after washing the cells)?

7) What is the export substrate for these transport assays (it cannot be IBB)?. Also, do they use similar a protocol in Figure 3 and Figure 7? Moreover, why do they use Triton-X100? Triton-X100 permeabilizes the nuclear envelope. If Triton-X100 is used, the authors need to include extensive control experiments to demonstrate that the nuclear envelope is intact in their assay system.

8) The authors need to show the evidence that their CYFP or NYHP tagged Ran is functional.

9) The authors claim "animal CRM1's H9 loop is far from NES groove, thus opening its NES groove for interaction with NES of RanBP1". The authors need to show the evidence for this claim.

10) The paper is very difficult to read. It is full of grammatical errors and needs extensive proofreading. Also, the authors start with a statement that they wanted to solve the X-ray structure of RanBP1 complexes but then they never did this or don't state why they didn't do this etc.

[Editors' note: further revisions were requested prior to acceptance, as described below.]

Thank you for resubmitting your article "Distinct RanBP1 nuclear export and cargo dissociation mechanisms between fungi and animals" for consideration by *eLife*. Your revised article has been reviewed by one peer reviewer, and the evaluation has been overseen by a Reviewing Editor and Ivan Dikic as the Senior Editor. The reviewers have opted to remain anonymous.

The reviewers have discussed the reviews with one another and the Reviewing Editor has drafted this decision to help you prepare a revised submission.

Whereas the reviewers still feel that the observations described in this manuscript are potentially of interest to the field, unfortunately, they were not satisfied by your revisions.

1) The choice of Triton X-100 for the in vitro transport assay remains problematic as it is expected to permeabilize the nuclear envelope and the provided supplementary figure is not sufficient to alleviate this concern. Why don't they use digitonin? For permeabilization with Triton X-100 additional control experiments are needed. That would include the addition of non-NPC permeable dyes to demonstrate that the nuclear envelope is fully intact.

2) The reviewers were concerned about the functionality of the N- and C-terminally tagged Ran variants. The provided pull-down experiment does not fully address this concern. The appropriate experiment would be to perform an endogenous Ran knockdown to test for rescue by the Ran-variants used in this study.

3) The interpretation of the import experiments examining the nuclear/cytoplasmic localization of expressed SV40 T-NLS or PKI-NES substrate in mammalian cells and yeast cells is difficult. The SV40 T NLS has a weaker binding activity for yeast importin α compared to mammalian importin. Also, although the PKI NES apparently display a similar binding activity with yeast CRM1 and mammalian CRM1 in Figure 8—figure supplement 1, binding was only examined at one protein concentration (this is not "affinity"). In addition, the expression levels of NES substrate versus endogenous CRM1/RanGTP proteins in yeast and mammalian cells should be examined.

---

## [Author Response]

Essential revisions:1) The paper largely describes the differences between the animal and yeast complexes but the physiological relevance of the findings remains unclear. The authors discuss some of the implications but it would be important to test at least some of these possibilities. Of note, by doubling the energy expenditure, animal cells should be able to do 'more work' and could export against a higher concentration difference. Is there any evidence that this is the case in animal cells?

We really thank the reviewer for this comment. We measured the nuclear-cytoplasmic ratio for transfected cargoes in yeast and human cells, and compared them. Interestingly, the same cargoes that predominantly localize to the cytoplasm and nucleus in animals cells are localized at much lower nucleo-cytoplasmic concentration difference, suggesting possibly more efficient nuclear transport system in animals (new Figure 8C), though these cargoes binds to transport receptors with similar affinity (new Figure 8—figure supplement 1). These are added as Figure 8C, D, and we added a few lines of discussion (subsection “Biological significances of forming trimeric or tetrameric complex”) and a new paragraph describing the obtained results in the subsection “Higher efficiency of nuclear export and import in mammals”.

2) On a similar note, can yeast cells lacking yCRM1 and yRanBP1 be complemented with the two animal counterparts (or vice versa)? Or does the triple mutant rescue the yeast deletion?

We thank the reviewer for this suggestion. We failed to obtain temperature sensitive yeast strains. However, we successfully performed knock down experiments of the endogenous CRM1 or RanBP1 in mammalian cells and rescues with either human or yeast constructs mentioned. Though yRanBP1 is an essential gene, human RanBP1 knock down and co-comitant transfection of various mammalian or yeast RanBP1 proteins (WT or mutants) did not affect cellular localization of CRM1, nuclear export cargo (NFκB) and nuclear import cargo (GFP-NLS), likely because that there is functional redundant gene RanBP2 in mammals (PMID: 27160050). This result also indicates that the expression level of RanBP1 is probably too low to disrupt cellular RanGTP gradient, in order to show any cargo localization defects. These results are added as Figure 7—figure supplement 1.

On the other hand, cytoplasmic localization of endogenous RanBP1 and NFκB was perturbed by siCRM1, which could be rescued by expression of either hCRM1 and yCRM1^T753Q^ but not yCRM1 and hCRM^1Q742T^ (new Figure 7E-G). It is quite surprising because hCRM1^Q742T^ is much more like hCRM1 than yCRM1^T753Q^ in sequence, however function-wise the opposite is true. This suggests that the mutation hCRM1Q742T (or yCRM1T753Q) play important roles in protein function. Negative control using transfected mCherry-NLS displayed no localization change under siCRM1 and different rescue conditions (new Figure 7H). These results were described in a new paragraph in the subsection “Identified mutations play important roles in RanBP1 localization and nuclear export of cargo”.

3) Why do both ΔNES and ΔC trigger cargo release, yet both these domains are required for efficient binding? What is the phenotype of ΔNES or ΔC mutants when expressed in tissue culture upon knockdown of the endogenous proteins?

We actually demonstrated that ΔNES and ΔC are neither essential (Figure 1E) nor sufficient (Figure 1C) for binding to CRM1-RanGTP (subsection “Mouse RanBP1’s NES is necessary for CRM1 binding and its nuclear export”).

Knock down endogenous RanBP1 and rescue with ΔNES and ΔC mutants mis-localized the protein to the nucleus (Figure 1F, Figure 7—figure supplement 1). As explained in (2), it did not affect the nuclear transport of cargo, probably due to the presence of functionally redundant RanBP2 and relatively low expression level.

*4) It is not satisfying that the authors write, we 'believe' that the NES doesn't contribute to dissociation vial competition (subsection “Mouse RBD dissociates cargo through sequestering RanGTP”). That needs to be more vigorously tested. Similarly, they write that the C-terminus is 'required' for CRM1 cargo dissociation. It would be correct to say it is sufficient, but they don't show that it is required (as ΔC still dissociates in vitro). As mentioned above (point 1) it would be important to test whether the C-terminus is needed* in vivo.

Thank you for this comment. We performed a more vigorous cargo dissociation experiment (Figure 2A) with a concentration gradient of different RanBP1s to support our argument. This is the new Figure 2—figure supplement 1. We also rewrote this paragraph as: “Pull down assay showed that ΔC or NESmut dissociated cargoes as potently as WT mRanBP1 (Figure 2A, Figure 2—figure supplement 1), suggesting that NES of mRanBP1 is dispensable for cargo dissociation. […] Taken together, we conclude that mRanBP1 does not dissociate cargo through direct competition of cargo’s NES.”

We actually wrote that “ΔC” is required, not “C terminus”. We replaced ‘ΔC’ with ‘the Ran binding domain’ to avoid confusion. Without C-terminus, the proteins is mis-localized to the nucleus. Please refer to our replies to point 2 and 3.

5) Why does the stable BIFC complex remain in the nucleus at steady state? If functional, it might be expected that the artificial Ran dimer might get stuck in the cytoplasmic after export? Is the BIF Ran complex functional and can it still shuttle? Also, the cells in Figure 5 (particularly the 3rd row) do not look very healthy and the results should be quantified.

Thanks for this comment. Only nucleus has high concentration of RanGTP, which is the prerequisite of RanBP1 tetramer formation. At steady state, probably a fraction of the tetramer is unexported (yet to be exported), which is observed. The transiently formed YFP would probably dissociate after Ran dimer dissociated from the complex in the cytoplasm, since we did not observe cytoplasmic YFP signal. As CRM1 is untouched, we believe that the complex could still shuttle through the NPC as other CRM1 complexes. The tetramer might have undiscovered function in the cytoplasm (or the nucleus). The tetramer should have short cytoplasmic life, and probably would be quickly dissembled, though a small fraction might be able to shuttle back to the nucleus before being dissembled. We modified/added more discussion as: “In the nucleus, animal RanBP1 cargo dissociation activity is inhibited with excessive RanGTP and it would not dissociate any CRM1 cargo or its own NES before reaching the cytoplasm. When animal RanBP1 tetrameric complex enters the cytoplasm, where RanBP1/RanGAP is present at high concentration and RanGTP at low concentration, the complex would be probably quickly dissembled (shutting off cytoplasmic YFP signal in the BiFC experiment).”

We repeated the whole experiment in Figure 5, replaced those figures with better quality figures and quantify those results as suggested.

In addition, we performed FRET analysis using florescent labelled Ran proteins to demonstrate tetramer formation in the nucleus. The data showed that FRET is stronger for cells transfected with RanBP1^Δlinker^-NLS than RanBP1^ΔNES^-NLS (new Figure 5—figure supplement 2), suggesting the existence of nuclear tetramer. A few sentences describing the data is added in the subsection “Detection of RanBP1 tetramer in the nucleus”.

6) The description of the in vitro nuclear export assay (in the Materials and methods) is not adequate and it remains unclear how these experiments were performed. Do they first perform nuclear import reactions, then a nuclear export reaction (after washing the cells)?

Thanks for this comment. The export is done after nuclear import and washing the cells. We rewrote this paragraph with reference to the previous studies and made further description/explanations in the subsection “In vitro nuclear export using semi-permeabilized cells”.

7) What is the export substrate for these transport assays (it cannot be IBB)?. Also, do they use similar a protocol in Figure 3 and Figure 7? Moreover, why do they use Triton-X100? Triton-X100 permeabilizes the nuclear envelope. If Triton-X100 is used, the authors need to include extensive control experiments to demonstrate that the nuclear envelope is intact in their assay system.

Thanks for the reviewer that pointed this out. The export substrate is GST-NES^PKI^-NLS^IBB^ but not GST-IBB. We corrected this error in the Materials and methods (subsection “In vitro nuclear export using semi-permeabilized cells”). Yes we used similar protocols in Figure 3 and Figure 7 (added in Figure 8 legend). The very low concentration of Triton-X is used to prevent non-specific cytoplasmic binding. We provided control images to prove that the nuclear envelope is intact (see Figure 3—figure supplement 1).

8) The authors need to show the evidence that their CYFP or NYHP tagged Ran is functional.

Thanks for suggesting this experiment. We performed a RanBP1 pull down and western blot experiment to show that the two tagged Ran proteins are functional (Figure 5—figure supplement 1).

9) The authors claim "animal CRM1's H9 loop is far from NES groove, thus opening its NES groove for interaction with NES of RanBP1". The authors need to show the evidence for this claim.

Sorry for this misleading sentence. It is known that when NES is bound, its H9 loop should be shifted away from NES groove to allow NES binding (PMID: 24631835). We rewrote these sentences as “In this model, unlike the yRanBP1 that contacts both yCRM1 and yRanGTP, animal RanBP1’s RBD only binds RanGTP (but not CRM1). Further, in contrast to H9 loop stabilized closure of NES groove in yeast (17), animal CRM1’s H9 loop would be shifted away from the NES groove (as in pdb 3NC0)(30), opening its NES groove for interaction with NES of RanBP1 (Figure 4A).”

10) The paper is very difficult to read. It is full of grammatical errors and needs extensive proofreading. Also, the authors start with a statement that they wanted to solve the X-ray structure of RanBP1 complexes but then they never did this or don't state why they didn't do this etc.

Thanks for this suggestion. We sent this manuscript to a professional language editor to correct the grammar errors and we made several proof readings. We attempted to solve the crystal structure, but we failed since that it did not form a good complex (Figure 1—figure supplement 2). We rewrote those sentences in the subsection “Mouse and yeast RanBP1 bind to CRM1-RanGTP differently”.

[Editors' note: further revisions were requested prior to acceptance, as described below.]

[…] Whereas the reviewers still feel that the observations described in this manuscript are potentially of interest to the field, unfortunately, they were not satisfied by your revisions.

*1) The choice of Triton X-100 for the* in vitro *transport assay remains problematic as it is expected to permeabilize the nuclear envelope and the provided supplementary figure is not sufficient to alleviate this concern. Why don't they use digitonin? For permeabilization with Triton X-100 additional control experiments are needed. That would include the addition of non-NPC permeable dyes to demonstrate that the nuclear envelope is fully intact.*

We used Digitonin to permeabilize the cell membrane. Low concentration of Triton-X100 was used to prevent non-specific binding. Triton-X is much cheaper than digitonin, and more stable than BSA and Digitonin when prepared in large stock on shelf. As the reviewer suggested, we now used anti-PARP antibody to test whether the nuclear envelope are permeabilized. The results showed that the envelope is intact in low Triton-X100 condition. (added in Figure 3—figure supplement 1) Together with previous data (nuclear import level is similar in the presence of BSA or low Triton-X), we hope it is acceptable that low concentration of Triton-X could be used as a buffer supplement for preventing non-specific binding. We thank the reviewer for suggesting this experiment.

2) The reviewers were concerned about the functionality of the N- and C-terminally tagged Ran variants. The provided pull-down experiment does not fully address this concern. The appropriate experiment would be to perform an endogenous Ran knockdown to test for rescue by the Ran-variants used in this study.

Thanks for suggesting this experiment. We tried to test Ran to look for differences in cargo nuclear export (hopefully this is what the reviewer requested). However, no statistical cargo distribution difference was found (Author response image 1, obviously the rescue experiments also showed similar localization). It is possible that Ran knock down impaired nuclear import and export to similar extent.

**Author response image 1. respfig1:** Nuclear export of NFκB in the presence of Ran knock down and rescues with Ran fusions.

In addition, we observed that the expressed Ran fusions is much less compared to the WT protein (Author response image 2). Thus, it may be hard to observe a rescue of cellular phenotypes.

**Author response image 2. respfig2:** Expression level (HeLa cells) of Ran fusions in comparison with endogenous Ran. The transfection efficiency for each plasmid was about 30%. The bands were stained with Ran antibody. The Ran fusion bands were verified by HA and MYC antibodies.

Since one of the most important functions of Ran is to bind to RanBP1, we devised a more sophisticated pull down assay to show that the two Ran fusion could rescue binding to GST-RanBP1 in the presence of Ran WT knock down (Figure 5—figure supplement 1, replaced the old figure). With these data and similar cellular localization pattern (Figure 5—figure supplement 3), we argue that the Ran variants are not mis-folded when expressed, and that their specific binding to RanBP1 made it possible for us to observe what was presented in Figure 5B, which suggest that Ran could form RanBP1 tetramer in the nucleus.

3) The interpretation of the import experiments examining the nuclear/cytoplasmic localization of expressed SV40 T-NLS or PKI-NES substrate in mammalian cells and yeast cells is difficult. The SV40 T NLS has a weaker binding activity for yeast importin α compared to mammalian importin. Also, although the PKI NES apparently display a similar binding activity with yeast CRM1 and mammalian CRM1 in Figure 8—figure supplement 1, binding was only examined at one protein concentration (this is not "affinity"). In addition, the expression levels of NES substrate versus endogenous CRM1/RanGTP proteins in yeast and mammalian cells should be examined.

We agree with the reviewer that the interpretation of the comparison between mammalian and yeast cells is difficult (subsection “Biological significances of forming trimeric or tetrameric complex”). We repeated the NES/NLS binding assays with more concentrations, which showed that the binding affinities are very comparable between yeast and human (Figure 8—figure supplement 1B). Indeed, SV40NLS binds to human importin α slightly stronger, which would imply more nucleus localization level in human compared to yeast for mCherry-NES. However, Figure 7C right panel showed the opposite, suggesting that the mCherry-NES localization difference was not due to binding strength difference. These were explain in the main text and Figure 8—figure supplement 1 legend.

We observed that the expression level of NES in yeast is rather low compared with mammalian cells (huge difference by western as in Figure 8—figure supplement 2A and by the fact that mCherry signal in yeast is hardly observable under microscope). Unfortunately, our human CRM1/Ran antibodies did not recognize yeast proteins (Figure 8—figure supplement 2B). Published reports showed that CRM1 and Ran are expressed at comparable levels in yeast and human cells (Figure 8—figure supplement 2C). Thus, the lower transport efficiency seen in yeast should not be due to saturating amount of expressed cargoes. This discussion is added to main text as: “In addition, mCherry-NES expression level was about 30 times less in yeast cells compared to mammalian cells (Figure 8—figure supplement 2A), while the Ran and CRM1 concentrations in both cells are similar (Figure 8—figure supplement 2B, C), suggesting that the lower export efficiency in yeast should not be due to saturation of export capacity.”